# Hinokiflavone as a Potential Antitumor Agent: From Pharmacology to Pharmaceutics

**DOI:** 10.3390/cells15010017

**Published:** 2025-12-22

**Authors:** Fengrui Liu, Ranyi Li, Xiaolei Zhou, Xiaoyu Li

**Affiliations:** 1Department of Pharmacy, Zhongshan Hospital, Fudan University, Shanghai 200032, China; 24211210082@m.fudan.edu.cn (F.L.); li.ranyi@zs-hospital.sh.cn (R.L.); 2College of Biological Science and Engineering, Fuzhou University, Fuzhou 350108, China

**Keywords:** hinokiflavone, multidrug resistance, anticancer mechanisms, tumor microenvironment, pharmacokinetics, nanodelivery systems

## Abstract

Hinokiflavone (HF), a natural C-O-C-linked biflavonoid originally isolated from *Chamaecyparis obtusa*, is a promising multifunctional antitumor agent. Despite challenges posed by multidrug resistance and tumor heterogeneity, HF demonstrates notable therapeutic potential through a multifaceted pharmacological profile. HF exerts broad-spectrum anticancer effects by targeting multiple oncogenic pathways, including the MDM2-p53 axis, MAPK/JNK/NF-κB signaling, ROS/JNK-mediated apoptosis, and Bcl-2/Bax-regulated mitochondrial pathways. These actions are further complemented by inhibition of cell proliferation through cell cycle arrest and suppression of metastasis via downregulation of matrix metalloproteinases and reversal of epithelial–mesenchymal transition. Additionally, HF displays antioxidant, anti-inflammatory, and antimicrobial activities, enhancing treatment efficacy. However, its clinical translation remains limited by poor aqueous solubility, low oral bioavailability, and incomplete pharmacokinetic characterization. Recent advances in nanotechnology-based formulation strategies, such as polymeric micelles and metal–organic frameworks, have enhanced HF’s bioavailability and in vivo antitumor efficacy. This review comprehensively delineates HF’s molecular mechanisms of anticancer action, evaluates its pharmacokinetics and bioformulation developments, and highlights challenges and prospects for clinical application. Integration of tumor microenvironment-responsive delivery systems with synergistic therapeutic strategies is essential to fully realize HF’s therapeutic potential, positioning it as a valuable scaffold for novel anticancer drug development.

## 1. Introduction

Malignant tumors represent one of the foremost global public health threats, with their associated disease burden continuing to escalate. According to the latest statistics from the International Agency for Research on Cancer, approximately 19.96 million new cancer cases and 9.74 million related deaths were reported worldwide in 2022, with lung, colorectal, and liver cancers constituting the three leading causes of cancer mortality [1]. Despite substantial advances in contemporary oncological therapies—including surgical resection, chemotherapy, radiotherapy, targeted therapy, and immunotherapy—numerous clinical challenges persist [2]. Multidrug resistance (MDR) is implicated directly or indirectly in approximately 80–90% of cancer-related deaths [3], notably diminishing therapeutic outcomes following conventional chemotherapy and representing a predominant cause of treatment failure [4,5]. Cancer is not a singular disease entity but a complex microecological system comprising heterogeneous cellular subpopulations exhibiting distinct genotypes and phenotypes [6]. Such tumor heterogeneity undermines the efficacy of monotherapeutic approaches and leads to substantial variability in individual patient responses to identical treatment regimens [7]. For instance, in HER2^+^ metastatic breast cancer, nearly 20% of patients fail to achieve objective remission despite standard regimens involving pertuzumab, trastuzumab, and docetaxel [8]. Similarly, although immunotherapies confer durable responses in certain patients, overall response rates remain suboptimal [9,10]. These multifactorial challenges constrain the broad application of existing therapeutic modalities. Consequently, the development of novel antitumor agents—particularly those derived from natural products with unique mechanisms of action—has become a critical strategy to overcome these clinical bottlenecks.

Natural products are renowned for their structural diversity and distinct bioactivities, serving as a vital reservoir for antitumor drug discovery. Many natural products or their derivatives have become cornerstones of modern cancer treatment. For instance, irinotecan, a camptothecin derivative, is a key component of first-line therapy for metastatic colorectal cancer (e.g., the FOLFOXIRI regimen) [11], while doxorubicin remains the standard first-line treatment for soft tissue sarcomas [12]. The successful development of chemotherapeutics such as paclitaxel and vincristine has further stimulated the exploration of bioactive compounds derived from plants and microorganisms. Flavonoids, a class of plant-derived polyphenolic secondary metabolites, have demonstrated prophylactic and therapeutic efficacy against various malignancies, including breast [13], prostate [14], and colorectal cancers [15]. Their multitarget regulatory properties attract considerable attention in antitumor drug development. Notably, flavonoids and their derivatives have achieved clinical translation by targeting key signaling pathways such as IL-6/JAK2/STAT3 [16], PI3K/AKT [17], and cell cycle regulators including cyclin-dependent kinase (CDK) 4 and CDK9. However, the clinical advancement of many flavonoid compounds remains hindered, as approximately 50% of novel candidates entering clinical trials are discontinued due to insufficient pharmacological activity [18]. Recent studies have demonstrated that 3′–8″-linked biflavonoids exhibit stronger antioxidant, enzyme-inhibitory, cytoprotective, and anticancer activities than their monomeric flavonoid counterparts [19]. In this context, biflavonoids have attracted increasing scientific attention due to their distinctive molecular architecture and their overall enhancement in pharmacological activities [20].

Biflavonoids are dimers of two monomeric flavonoids, formed via C-C [21] or C-O-C [22] linkages, which extend the molecular conjugation system and augment hydrogen bonding sites. Compared to monomeric flavonoids, the increased number of phenolic hydroxyl groups in biflavonoids enhances enzyme inhibitory activity [23], and the dimeric structure optimizes spatial conformation, improving target binding affinity and markedly elevating pharmacological potency across various biological contexts, from enzyme inhibition to anticancer activity [20,24,25,26,27]. For instance, a recent in silico study demonstrated that dimeric flavonoids like hinokiflavone (HF) exhibit stronger binding than apigenin to PIM1 kinase, an oncogenic target overexpressed in glioblastoma, exemplifying this “dimer effect” in an anticancer setting [20]. This synergistic “dimer effect” often surpasses the activity increment attributable solely to increased hydroxyl groups. The structure-activity relationships (SAR) of C-C-linked biflavonoids have been investigated [28], with connectivity patterns and representative compounds illustrated in Figure 1. Retention of free hydroxyl groups at specific positions—such as C4′, C4‴, and C7—is critical for bioactivity enhancement. For example, sequential methylation of hydroxyl groups at C7, C4‴, C7″, and C4′ of cupressuflavone significantly diminishes anticancer efficacy against MCF-7 breast cancer cells, with half maximal inhibitory concentration (IC_50_) values increasing from 3.40 ± 0.3 to 397.89 ± 28.6 μM [29]. Similarly, methylation at C7 of amentoflavone (AF) substantially reduces its tumor-suppressive effects [30]. Lin et al. reported that the presence of an unsaturated bond between C2 and C3 within the pyran ring, alongside free phenolic hydroxyl groups, is essential for biflavonoid inhibition of HIV-1 reverse transcriptase [31]. Additionally, biflavonoid bioactivity correlates strongly with the mode of linkage [28]; for instance, C8-C8″ and C8-C3‴ linked biflavonoids display selective and potent antiproliferative effects across diverse cancer cell types. Sotetsuflavone, linked via C8-C3‴, inhibits dengue virus RNA-dependent RNA polymerase (RdRp) more effectively than the C6-C3‴ linked robustaflavone [32]. Biflavonoids exhibit antitumor effects through multiple synergistic mechanisms including apoptosis induction [6], cell cycle arrest [33], modulation of key signaling pathways [34,35,36], anti-angiogenesis [37], and inhibition of tumor migration and invasion [38], thereby interfering with tumor initiation, proliferation, and metastasis. Representative biflavonoids like cupressuflavone, robustaflavone, and agathisflavone have demonstrated therapeutic potential against breast [39], prostate [40], and glioblastoma cancers [41]. Given their structural advantages and superior bioactivity, biflavonoids offer a potential direction for improving the clinical translation efficiency of flavonoid-based therapies.

Among biflavonoids, HF has attracted considerable attention due to its distinctive molecular architecture and multitargeted pharmacological profile. In recent years, several insightful reviews have contributed to our understanding of HF. The work by Goossens et al. (2021) provided a foundational overview of C–O–C-type biflavonoids, focusing on HF’s general anticancer properties and early mechanisms, notably its role as a sentrin-specific protease 1 (SENP1) inhibitor and modulator of the extracellular signal-regulated kinase (ERK)/nuclear factor kappa-light-chain-enhancer of activated B cells (NF-κB) pathway [22]. Subsequently, Patel (2024) expanded the perspective, cataloging the broad biological potential of HF across various human diseases, including cancer [42]. This review provides a significant update by synthesizing the most recent discoveries of HF’s mechanisms (e.g., MDM2-p53 axis, casein kinase 2 (CK2) inhibition) and distinctly focuses on critically analyzing controversial pharmacological data, providing an in-depth discussion of pharmacokinetic challenges and advanced nano-formulation strategies, and offering a forward-looking perspective on translational barriers—aspects not comprehensively covered in earlier works.

HF is a prototypical C-O-C linked biflavonoid ether of apigenin units, originally isolated in 1958 from the *Chamaecyparis obtusa* [43,44]. It possesses a molecular formula of C_30_H_18_O_10_ and a molecular weight of 538.5 Da. Early structural characterizations mistakenly assigned the linkage as C4′-O-C8″ ether; this was later corrected by Nakazawa et al. to a C4′-O-C6″ connectivity, representing a 4′,6″-O-Biapigenin isomer [45,46], with multiple phenolic hydroxyl groups present (Figure 2). Unlike the rigid planar structures of C-C linked biflavonoids such as AF, the C-O-C linkage confers greater conformational flexibility to HF, enhancing its binding affinity to biomacromolecules [24]. At ambient temperature, HF is present as pale-yellow crystals or amorphous powder, has a melting point of approximately 253–258 °C, is sparingly soluble in water, but readily dissolves in methanol, ethanol, and DMSO [47]. HF is prone to degradation under acidic or photolytic conditions and should be stored protected from light [48]. SwissADME predictions estimate HF’s bioavailability at approximately 55% [49].

HF is widely distributed in various plant species beyond *Chamaecyparis obtusa*, including *Platycladus orientalis* [50], *Juniperus drupacea* [51], *Juniperus rigida* [52], *Podocarpus imbricatus* [53], *Selaginella tamariscina* [54], *Selaginella sinensis* [55], *Toxicodendron succedaneum* [56], *Cycas revoluta* [57], and *Cryptomeria japonica* [58]. Chemical synthesis of HF is facilitated by Ullmann condensation or Wessely–Moser rearrangement reactions [45,59,60], enabling scalable production. Preliminary studies revealed HF’s multifaceted bioactivities including anti-inflammatory [61], antiviral [25,31], and antioxidant effects [62]. In oncology, Lin et al. first reported HF’s potent cytotoxicity in 1989, showing an ED_50_ of 2.0 μg/mL against the oral squamous carcinoma KB cell line (later confirmed to be a HeLa derivative), identifying the ether-linked apigenin dimer structure as essential for cytotoxic activity [56].

HF exhibits notable antitumor mechanisms and advantages: compared to conventional chemotherapeutics such as cisplatin and doxorubicin, HF demonstrates tumor cell selectivity [63,64,65,66]. Its mechanisms involve multifaceted pathway modulations, including (i) regulation of the MDM2/p53 axis [63,67]; (ii) induction of apoptosis via ROS/JNK/caspase cascades [64]; and (iii) interference with MAPK/JNK/NF-κB signaling [64,68]. Of particular note, HF suppresses the activity of SENP1 by modulating the precursor mRNA splicing process of SENP1 [69]. SENP1 is aberrantly overexpressed in various malignancies and promotes tumor progression through the regulation of hypoxia-inducible factor-1-α-dependent angiogenesis [70,71]. Furthermore, HF reverses cisplatin resistance in bladder cancer cells [72], providing novel insights into overcoming clinical MDR.

Despite its promising antitumor potential, existing research on HF faces critical challenges: (1) insufficient systemic understanding of its multitarget interaction networks; (2) incomplete elucidation of SAR; (3) limited data on in vivo pharmacodynamics and pharmacokinetics (PK); and (4) lagging development of novel drug delivery systems. In response to these hurdles, this review will primarily focus on: (i) a comprehensive panorama of HF’s molecular antitumor mechanisms; (ii) strategies for activity enhancement via structural modification; (iii) innovative technologies to improve bioavailability; and (iv) prospects for combinatorial therapeutic regimens. We provide a significant update by integrating the most recent discoveries, such as HF’s modulation of the MDM2-p53 axis and its function as a CK2 inhibitor. More importantly, our manuscript addresses critical gaps left by previous summaries by: (i) offering a dedicated and critical analysis of inconsistent pharmacological data across studies; (ii) delivering a comprehensive and in-depth discussion on the pharmaceutical challenges (e.g., poor solubility, extensive metabolism) and the cutting-edge nano-formulation strategies designed to overcome them; and (iii) concluding with a rigorous, forward-looking assessment of specific translational barriers and requisite preclinical studies. By focusing on these underexplored yet crucial aspects, this review aims to serve as an indispensable guide for advancing HF from a promising lead compound towards a viable clinical candidate.

## 2. Methodology

To provide a comprehensive overview of the pharmacological potential and pharmaceutical development of HF, a systematic literature search was conducted using major scientific databases, including PubMed, Web of Science, Scopus, and Google Scholar. The search covered the period from the compound’s initial identification up to August 2025.

The search strategy employed a combination of the specific Mesh terms and free-text keywords: “Hinokiflavone”, “biflavonoids”, “anticancer” (OR “antitumor”, “cytotoxicity”), “pharmacokinetics” (OR “bioavailability”, “metabolism”), and “drug delivery” (OR “formulation”).

Selection Criteria: The literature selection followed specific inclusion and exclusion criteria to ensure the scientific quality of this review.

Inclusion Criteria: (1) Peer-reviewed original research articles and reviews published in English, as well as seminal works in other languages (specifically Japanese) containing primary data on HF isolation or characterization; (2) Studies explicitly investigating the isolation, chemical synthesis, molecular mechanisms, PK, or formulation strategies of HF; (3) Comparative studies involving HF and other natural product.

Exclusion Criteria: (1) Conference abstracts, editorials, and articles in languages other than English or Japanese that lacked accessible abstracts or data; (2) Studies lacking sufficient experimental data or clear methodological descriptions; (3) Articles where HF was part of a crude extract without specific characterization of its individual contribution.

All retrieved titles and abstracts were independently screened for relevance. The reference lists of selected articles were also manually checked to identify further relevant studies. This process ensured that the review synthesizes the most current and reliable evidence regarding the therapeutic prospects of HF.

## 3. Pharmacological Mechanisms of HF’s Anticancer Effects

Having outlined the structural uniqueness and natural sources of HF, we now delve into the core of its therapeutic potential: a multi-targeted pharmacological profile that acts on both the tumor cells and their surrounding microenvironment. HF exerts tumor-suppressive actions through a synergistic network of pathways, with its biflavonoid scaffold enabling simultaneous modulation of multiple cancer cell-intrinsic and microenvironmental targets (Figure 3). HF primarily acts directly on tumor cells, precisely interfering with malignant tumor behaviors by regulating critical processes including apoptosis induction, cell cycle arrest, and inhibition of invasion and metastasis. Additionally, HF modulates the tumor microenvironment (TME) through its potent antioxidant and anti-inflammatory activities, which eliminate potential carcinogenic risk factors and serve a cancer-preventive role. This dual mechanism of direct cytotoxicity and microenvironment modulation suggests that HF may exert relatively lower cytotoxicity toward normal cells compared with its effects on cancer cells, as observed in in vitro models, implying a potential for selective anticancer activity. However, these observations remain preliminary and require further in vivo validation. The following subsection systematically elaborates on the pharmacological mechanisms underlying HF’s antitumor activity.

### 3.1. Induction of Apoptosis

To understand the core of HF’s therapeutic efficacy, we first evaluate its direct cytotoxic mechanisms, starting with the induction of programmed cell death. Apoptosis, a gene-regulated programmed cell death mechanism, is essential for normal development and tissue homeostasis in multicellular organisms [73]. It ensures orderly removal of cells without triggering inflammation, in contrast to necrosis, thereby maintaining tissue microenvironment stability [74]. Cancer cells frequently suppress apoptotic pathways to gain immortality. The Bcl-2 protein family comprises key regulators of mitochondrial apoptosis, divided into three groups based on structure and function: anti-apoptotic Bcl-2 subfamily, pro-apoptotic Bax subfamily, and BH3-only proteins. In normal cells, the expression and activity of Bcl-2 and Bax are tightly regulated; together, they control mitochondrial membrane permeability, thereby, governing cytochrome c release and activation of caspase-3, -8, and -9, thus mediating intrinsic apoptosis [75]. The Bcl-2 targeting agent venetoclax is clinically approved for acute myeloid leukemia in elderly or chemotherapy-ineligible patients [76]. Disruption of the balance between apoptosis induction and inhibition equilibrium enables resistance to death signals, potentially leading to malignant transformation [77].

Tumor suppressor protein p53 functions as a transcription factor that halts the cell cycle and triggers apoptotic programs under stress [78]. MDM2 is a principal negative regulator of p53, attenuating its tumor suppressor function through dual mechanisms: as an E3 ubiquitin ligase, it targets p53 for proteasomal degradation [79], and it also inhibits p53’s transcriptional activity [80]. Mutations or deletions in p53 occur in nearly 50% of cancers [81], while MDM2 amplification correlates clinically with poor prognosis in dedifferentiated liposarcoma and breast cancer. No p53 activators have yet reached clinical application [82], rendering the p53-MDM2 axis an attractive target in oncology.

HF induces apoptosis through an integrated multi-layered signaling network. The following subsections delineate these mechanisms in detail, organized by regulatory nodes. Ilic et al. utilized virtual screening and molecular docking to predict HF binding near the MDM2-MDMX RING domain dimer interface. Biolayer interferometry confirmed HF’s binding affinity to the MDM2-MDMX RING dimer at 12 μM. In vitro ubiquitination assays demonstrated HF’s inhibitory effect on MDM2 auto-ubiquitination and, thus, E3 ligase activity. Mechanistically, HF reduces p53 ubiquitination and degradation, activates p53 signaling, and promotes transcription of downstream target genes, inducing apoptosis via both p53-dependent and -independent pathways. Cellular assays showed dose-dependent cytotoxicity against human leukemia AML-2 and HL-60, osteosarcoma U2OS, breast cancer MCF-7, and colon cancer HCT116 cells, while exerting minimal toxicity on BJ human fibroblast. Sensitivity correlated with p53 status; wild-type p53-expressing cells were more susceptible than p53-null lines. Notably, HF potently inhibited p53 wild-type AML-2 cells (IC_50_ = 4.93 ± 1.16 μM) [63]. Zhang et al. explored HF’s suppression of MDM2 expression using cycloheximide and MG132; results indicated HF did not alter MDM2 degradation but dose- and time-dependently inhibited *MDM2* mRNA synthesis, relieving p53 repression and activating apoptosis in HCT116 colon cancer cells. Molecular docking and dynamics suggested HF may bind ESR1, a key protein in RNA transcription, influencing *MDM2* mRNA expression, although upstream mechanisms require further experimental validation [67].

HF modulates Bcl-2/Bax expression to facilitate caspase-dependent apoptosis in diverse cancer models. In an in vivo study, Huang et al. administered HF at doses of 20 or 40 mg/kg to MDA-MB-231 breast tumor-bearing mice over 21 days, resulting in a significant reduction in tumor weight alongside a marked decrease in Ki-67-positive proliferating cells, thus confirming HF’s antitumor efficacy and its impact on cell proliferation within the TME. Complementing these in vivo findings, in vitro treatment of MDA-MB-231 cells with HF for 24 h dose-dependently decreased Bcl-2 levels while upregulating Bax expression, thereby enhancing apoptosis through activation of the mitochondrial pathway [83]. Similarly, Yang et al. reported HF-induced apoptosis in melanoma A375 cells characterized by increased Bax, decreased Bcl-2, dose-dependent cleavage of caspase-3, elevated intracellular ROS, and decreased mitochondrial membrane potential, confirming ROS-mediated mitochondrial apoptosis [65].

HF also regulates upstream signaling related to Bcl-2/Bax. The anti-tumor efficacy of HF has been rigorously demonstrated in vivo in colorectal cancer models. In the study by Zhou et al., HF treatment significantly suppressed tumor growth in nude mice bearing HCT116 colorectal cancer xenografts, with dose-dependent reductions in both tumor volume and weight compared to control animals. Mechanistically, HF administration in vivo resulted in increased apoptotic cell counts within tumor tissues, elevated Bax/Bcl-2 ratio, and enhanced caspase-3 activation, directly confirming mitochondrial apoptosis induction at the tissue level. These in vivo results are consistent with in vitro findings that HF induces potent, ROS-mediated apoptosis in colorectal cancer cells through modulation of the JNK and p38 MAPK pathways and inhibition of NF-κB activity [84]. In liver cancer cells, HF dose-dependently induces mitochondrial ROS accumulation, activating JNK and p38 MAPK, and JNK inhibition by SP600125 has been shown to rescue HF-induced apoptosis, restore the Bcl-2/Bax ratio, and downregulate cleaved caspase-3 [64]. Similarly, in chronic myelogenous leukemia K562 cells, HF activates the JNK/p38 MAPK axis, suppresses NF-κB, and induces caspase-dependent death. Notably, HF also promotes autophagy, as evidenced by increased LC3-II and decreased p62 expression, with partial rescue by the autophagy inhibitor chloroquine, suggesting HF-induced autophagy may serve a cytoprotective role [68]. Additionally, computational studies predict that HF inhibits PIM1 kinase, an oncogenic serine/threonine kinase, further supporting its pro-apoptotic and anti-tumor potential [20].

In summary, HF induces apoptosis through a multi-target, multi-pathway network predominantly centered on apoptosis signaling regulation. It stabilizes and activates p53 by inhibiting MDM2, modulates Bcl-2 family proteins to promote mitochondrial apoptosis, activates pro-apoptotic ROS/JNK/p38 MAPK signals, and suppresses survival pathways such as NF-κB signaling, culminating in caspase cascade activation. This multi-layered regulatory mechanism has been repeatedly validated in diverse in vitro and in vivo tumor models, establishing HF as a promising broad-spectrum pro-apoptotic agent.

### 3.2. Cell Cycle Arrest

In addition to triggering apoptosis, HF effectively halts uncontrolled tumor proliferation by inducing cell cycle arrest at various checkpoints, further impeding cancer growth. Cell cycle dysregulation is one of the hallmarks of tumorigenesis [2]. A defining feature of cancer cells is uncontrolled proliferation, driven by mechanisms that promote relentless cell division and tumor formation. As elaborated in Section 3.1, HF activates p53, which functions not only to induce apoptosis but also to enforce cell cycle checkpoints via transcriptional upregulation of CDK inhibitors such as p21. This section focuses on the cell type-specific patterns of HF-induced cell cycle arrest and the downstream effector proteins involved. The cell cycle comprises multiple phases: G0/G1 (pre-DNA synthesis), S (DNA synthesis), and G2/M (post-DNA synthesis/mitosis) [85]. The progression of cell division is regulated by a series of complex proteins, with the core regulators being the complexes formed by CDK and cyclins [86]. Specifically, Cyclin D1-CDK4/6 regulates G1 phase, Cyclin E-CDK2 controls the G1/S transition, and Cyclin B1/Cdc2 (CDK1) governs G2/M phase entry. CDK4/6 inhibitors such as ribociclib and abemaciclib have revolutionized treatment of hormone receptor-positive advanced breast cancer by selectively arresting cells at G1 phase and significantly prolonging patient survival [87].

The ability of HF to induce cell cycle arrest has been demonstrated both in vivo and across multiple cancer cell types in vitro, highlighting its broad antiproliferative potential. In hepatocellular carcinoma (HCC), Mu et al. showed that HF significantly suppressed tumor growth in SMMC-7721 xenograft-bearing nude mice. Mechanistically, HF administration upregulated phosphorylated p53 (Ser15) and p21 within tumor tissues, while downregulating key cell cycle regulators such as cyclin D1, CDK4, and CDK6, providing in vivo evidence for activation of the p53/p21 axis and G0/G1 phase arrest. These molecular alterations correlated with reduced tumor proliferation (decreased Ki-67 staining) and increased apoptosis (TUNEL positivity) in tumor sections, confirming that cell cycle blockade is an integral part of HF’s antiproliferative action in vivo [64].

In agreement with these animal model findings, in vitro studies further reveal the context-dependent nature of HF-induced cell cycle arrest. In SMMC-7721 and HepG2 HCC cell lines, HF exposure resulted in time- and dose-dependent inhibition of viability, G0/G1 phase accumulation, and similar changes in cell cycle protein expression as observed in vivo [64]. In contrast, HF induces cell cycle arrest at distinct checkpoints in other cancer models. In chronic myelogenous leukemia K562 cells and HCT116 colon cancer cells, G2/M phase accumulation is observed; the latter also exhibits p21 and 14-3-3σ upregulation and alleviation of MDM2-mediated p53 repression [67,68]. In melanoma cell lines A375 and B16, HF triggers S phase arrest, as demonstrated by flow cytometry and EdU incorporation assays [63,65].

In summary, HF exerts broad-spectrum antiproliferative activity via induction of cell cycle arrest, with phase specificity dependent on cell type. It induces G0/G1 arrest in HCC, S phase arrest in melanoma, and G2/M arrest in chronic myelogenous leukemia and colon cancer cells. Mechanistically, HF frequently modulates the p53-p21 signaling axis, activating p53 and upregulating the CDK inhibitor p21, which in turn modulates downstream proteins such as Cyclin D1, CDK4/6, or Cdc2, thereby inhibiting the cell cycle. It is noteworthy that HF’s cytotoxicity arises from synergistic mechanisms including apoptosis induction, cell cycle arrest, autophagy induction, and modulation of key survival pathways, with differential efficacy across cancer types (Table 1). Currently, most research focuses on in vitro experiments, necessitating further validation in animal models to establish in vivo therapeutic potential.

### 3.3. Inhibition of Tumor Metastasis

Having explored how HF suppresses tumor growth and survival, we now examine its critical role in inhibiting metastasis—the primary cause of cancer mortality—by targeting cell migration, invasion, and the epithelial–mesenchymal transition (EMT). HF has shown convincing antimetastatic activity supported by both in vivo and in vitro evidence across diverse tumor models. 

In vivo, HF administration significantly reduced metastatic potential in xenograft models of both colorectal and breast cancer [83,84]. Immunohistochemical analyses demonstrated markedly decreased matrix metalloproteinase (MMP) 2 and MMP9 positive areas in tumor tissues from HF-treated mice compared to controls, indicating profound suppression of matrix remodeling enzymes that are critical for invasion and metastasis [83,84]. Consistent with these in vivo findings, in vitro studies revealed that HF downregulates the expression of MMP2 and MMP9 and upregulates tissue inhibitor of metalloproteinases (TIMP2), thereby restoring the MMP/TIMP balance and dose-dependently impeding migration and invasion of colorectal cancer cells [84]. Similarly, in melanoma models, treatment of A375 cells with HF led to significant inhibition of migratory capacity, as evidenced by wound healing and transwell assays, accompanied by reduced MMP2 and MMP9 protein levels [65].

Furthermore, HF suppresses metastasis by interfering with EMT, a crucial biological process enabling tumor cells to acquire migratory and invasive phenotypes, characterized by downregulation of epithelial markers such as E-cadherin and upregulation of mesenchymal markers including N-cadherin and vimentin [88]. In breast cancer cells treated with HF, a dose-dependent increase in E-cadherin and decrease in N-cadherin expression was observed by Western blot, consistent with EMT reversal or inhibition, culminating in suppressed migration and invasion [83]. These mechanistic effects were corroborated in tumor xenografts by immunohistochemistry, highlighting the in vivo relevance of HF-induced EMT inhibition [83].

HF mediates its anti-metastatic activity primarily through downregulation of MMPs expression and inhibition of EMT processes, forming a molecular basis for tumor metastasis suppression. However, current in vivo evidence relies heavily on migration/invasion marker evaluation, and direct functional validation of HF’s antimetastatic effects and mechanisms remains an imperative area for further investigation.

HF exerts robust direct antitumor activity via a multi-targeted network. It functions not only through modulation of the p53-MDM2 axis and mitochondrial apoptosis pathways, but also by enforcing cell cycle arrest at specific checkpoints across diverse tumor cell lines, and suppressing invasion and metastasis via the inhibition of EMT and downregulation of MMPs. However, solid tumors are not isolated cell masses but rather complex ecosystems where progression is critically dependent on interactions between tumor cells and the surrounding microenvironment. Beyond the direct cytotoxicity described above, HF demonstrates the potential to modulate the TME, complementing its anticancer efficacy by ameliorating oxidative stress and inflammatory conditions, as will be elaborated in the following section.

### 3.4. Antioxidant Effects

The antitumor efficacy of HF extends beyond direct cytotoxicity. A foundational aspect of its activity lies in its potent antioxidant capacity, which helps mitigate the oxidative stress that fuels tumor initiation and progression. ROS is a collective term for oxygen-containing free radicals generated as byproducts of cellular metabolism, mainly including superoxide anions, hydrogen peroxide, and hydroxyl radicals. Under physiological conditions, intracellular ROS homeostasis is maintained by antioxidant defense systems (e.g., superoxide dismutase (SOD), catalase, and glutathione peroxidase), which in turn preserves redox balance [89]. Low ROS levels function as pivotal secondary messengers in cellular signaling pathways regulating proliferation, differentiation, and immune responses [90]. However, excessive ROS production that overwhelms endogenous antioxidant capacity leads to oxidative stress, which is closely associated with the initiation and progression of cancer, aging, and other diseases [91].

Oxidative stress constitutes a critical driver of oncogenesis, tumor progression, and therapeutic resistance. Excess ROS induces DNA damage and genomic instability, which can activate oncogenes or cause loss-of-function mutations in tumor suppressor genes via epigenetic or mutational mechanisms, thereby initiating carcinogenesis [92]. Additionally, ROS serve as signaling molecules that irreversibly oxidize cysteine residues within protein kinases and phosphatases, leading to persistent activation of pro-oncogenic signaling pathways such as ERK, PI3K/AKT, and NF-κB. These pathways propagate proliferation signals and promote tumor cell survival [93,94,95].

Considering the tumor-promoting role of ROS, dietary antioxidants have been employed to restore redox homeostasis and mitigate cancer risk [96,97]. Traditional medicinal plants, such as *Selaginella* and *Ginkgo biloba*, are widely utilized in managing chronic diseases owing to their antioxidant constituents. For example, Further evaluations of *Selaginella sinensis* ethyl acetate extracts using High-Performance Liquid Chromatography- 2,2-diphenyl-1-picrylhydrazyl (HPLC-DPPH) demonstrated that quercetin had a DPPH IC_50_ of 3.2 ± 0.02 μM, outperforming rutin (3.8 ± 0.03 μM), while apigenin, AF, and HF exhibited moderate activity at higher concentrations (75 μM) [98]. DPPH-Ultra-High Performance Liquid Chromatography-Q-Time-of-Flight Mass Spectrometry (DPPH-UPLC-Q-TOF/MS) analysis of biflavonoids in *Selaginella doederleinii* ethyl acetate extracts highlighted HF as the most potent radical scavenger among the biflavonoids tested, followed by podocarpusflavone A, bilobetin, and ginkgetin [62].

Beyond dietary antioxidants, specific natural compounds such as HF have demonstrated intrinsic free radical-scavenging capacity. Using DPPH-UPLC-Q-TOF/MS screening, Wang et al. demonstrated that purified HF exhibited the most potent radical-scavenging activity among biflavonoids tested, including podocarpusflavone A, bilobetin, and ginkgetin [62]. Target-guided isolation by offline DPPH-HPLC followed by high-speed countercurrent chromatography has corroborated these screening results and yielded purified HF for direct bioassay; in those targeted assays, HF displayed measurable DPPH scavenging (reported at 75 μM), whereas small-molecule monoflavonoids such as quercetin exhibited lower IC_50_ values in the same screening framework (e.g., quercetin IC_50_ ≈ 3.2 μM) [98]. Critically, HF’s antioxidant effects extend beyond in vitro radical scavenging to encompass redox-modulating activity in biological systems. In a high-fat diet-induced rat model, purified HF (25–50 mg/kg) significantly attenuated oxidative stress by activating the EGFR/PI3K/Akt/eNOS signaling axis, enhancing nitric oxide bioavailability and upregulating SOD1 and catalase expression, collectively reducing hepatic ROS and lipid peroxidation [99]. These complementary lines of evidence-(i) identification of HF as a dominant radical scavenger and (ii) in vivo attenuation of oxidative stress through redox-modulating signaling—establish that HF possesses biologically and therapeutically relevant antioxidant activity in the context of chronic disease prevention and TME modulation.

The antioxidant capacity of HF also relates directly to its hepatoprotective effects, as demonstrated by multiple models. Alqasoumi et al. isolated five diterpenes and four flavonoid derivatives including HF and cupressuflavone from *Juniperus phoenicea*. Among the isolated compounds, HF exhibited the most pronounced hepatoprotective activity in a carbon tetrachloride (CCl_4_)-induced liver injury rat model, significantly reducing serum levels of alanine aminotransferase (ALT, also known as serum glutamic-pyruvic transaminase), aspartate aminotransferase (AST, also known as serum glutamic-oxaloacetic transaminase), and total bilirubin to levels comparable with the standard hepatoprotective agent silymarin. Histopathological examination of liver tissue further corroborated HF’s efficacy, revealing restoration of normal hepatic architecture and reduced necrotic areas, mirroring the protective effects observed with silymarin treatment [100]. However, in another study, the combination of HF and glycyrrhizin provided better protection than either agent alone but fails to surpass silymarin, as assessed in the same CCl_4_-induced liver injury rat model [101], suggesting discrepancies requiring further validation. Liu et al. demonstrated HF’s hepatoprotective effects against acetaminophen (APAP)-induced liver injury in both animal models and cell culture systems. In a murine APAP-induced acute hepatic injury model, HF treatment dose-dependently reduced serum ALT levels and significantly improved hepatic histopathology, with reduced centrilobular necrosis, inflammatory infiltration, and improved liver structure compared to APAP-only controls. Mechanistic investigations in HepG2 cells revealed that HF significantly alleviated APAP-induced cytotoxicity by restoring malondialdehyde (MDA), glutathione (GSH), and SOD levels, thereby re-establishing cellular redox homeostasis. Importantly, findings from cell culture were corroborated in vivo: Western blot analysis of liver tissue from HF-treated mice confirmed suppression of apoptotic markers (reversal of pro-apoptotic Bax upregulation and anti-apoptotic Bcl-2 downregulation) and pyroptotic markers (reduced NLRP3 inflammasome activation and gasdermin D N-terminal fragment expression), consistent with in vitro observations in HepG2 cells. Immunofluorescence staining of liver sections further validated effective inhibition of both apoptosis and pyroptosis by HF treatment in vivo. Mechanistically, HF dose-dependently activated the SIX4/AKT/STAT3 signaling axis both in vitro and in vivo; siRNA-mediated SIX4 knockdown or pharmacological STAT3 inhibition (S3I-201) in HepG2 cells markedly attenuated HF’s protective effects, while immunohistochemical analysis of liver tissue from HF-treated mice demonstrated increased phospho-STAT3 nuclear translocation, underscoring this pathway’s centrality in HF’s hepatoprotective mechanism. Notably, HF pretreatment inhibited APAP-stimulated ROS accumulation and prevented mitochondrial membrane potential collapse specifically in vitro, with in vivo confirmation of these endpoints pending [102].

Although HF’s intrinsic in vitro free radical scavenging ability and in vivo antioxidant activity have been well demonstrated, most antioxidant assessments rely heavily on the DPPH assay, which inadequately reflects biological system complexity. Therefore, the direct contribution of HF’s antioxidant activity to cancer prevention warrants further investigation. Given the frequent progression from liver injury through hepatitis, cirrhosis, to HCC [103], HF’s hepatoprotective efficacy suggests potential in reversing inflammation-driven carcinogenesis. It is noteworthy that HF exhibits context-dependent ROS modulation: in normal or stressed hepatocytes, HF acts as an antioxidant to suppress excessive ROS accumulation and maintain redox homeostasis; conversely, in tumor cells (as detailed in Section 3.1), HF induces ROS generation to trigger apoptotic cascades, demonstrating its dual regulatory capacity. Additionally, while HF exerts protection via SIX4/AKT/STAT3 activation, this pathway is also implicated in promoting tumor progression across various cancer types [104]. Future studies must delineate HF’s tumor cell-specific effects, dose–response relationships, and therapeutic windows to facilitate its translation into clinical practice.

### 3.5. Antimicrobial and Anti-Inflammatory Effects

Beyond countering oxidative stress, HF also exerts significant antimicrobial and anti-inflammatory effects. Chronic inflammation is a recognized oncogenic factor, sustaining a pro-tumorigenic microenvironment characterized by infiltrating inflammatory cells, cytokines, growth factors, and elevated ROS levels, which collectively promote malignant proliferation, angiogenesis, and resistance to cell death [105]. Persistent activation of inflammatory mediators such as TNF-α, IL-1β and signaling pathways like NF-κB disrupts normal DNA repair and impairs antitumor immune surveillance [106]. Additionally, certain pathogens act as direct carcinogens [107]; viruses such as human papillomavirus (HPV), Epstein–Barr virus (EBV), and hepatitis B virus (HBV) encode proteins that interfere with host cell cycle regulation and tumor suppressor pathways [108], while bacteria like *Helicobacter pylori* promote cancer indirectly by inducing chronic inflammation that fosters a tumor-supportive immune milieu [109]. Therefore, effective suppression of chronic inflammation and clearance of oncogenic pathogens constitute crucial adjunct strategies in comprehensive cancer management. Natural products with broad-spectrum antimicrobial and anti-inflammatory properties thus show distinct potential in the field of cancer research.

The investigation into HF’s antiviral properties is highly relevant to its anticancer potential, as several viruses are established oncogenic pathogens. The ability to inhibit such viruses directly contributes to a cancer chemopreventive strategy. Using radiolabeled assays, Coulerie et al. evaluated biflavonoids from *Dacrydium balansae* against dengue virus polymerases, revealing that HF possessed the strongest inhibition of DV-NS5 RdRp and DV-NS5 protein with IC_50_ values of 0.26 μM and 0.75 μM, respectively. HF also showed the most potent inhibition of West Nile virus NS5 RNA polymerase [25]. Konoshima et al. assessed EBV early antigen (EBV-EA) activation inhibition in Raji cells treated with various biflavonoids and flavonoids; kayaflavone, sotetsuflavone, and apigenin were most potent, while HF exhibited significant inhibition at high cell viability levels (viral activation reduced to 65.3% at 3.2 nM and completely suppressed at 32 nM) [110]. Notably, EBV is a well-known oncovirus linked to lymphomas and nasopharyngeal carcinoma, positioning HF’s activity here as a potential preventive mechanism. Lin et al. found HF effectively inhibited HIV-1 reverse transcriptase (IC_50_ = 62 μM) and showed antiviral activity in peripheral blood mononuclear cells (IC_50_ = 4.1 μM), though high cytotoxicity (EC_50_ = 9 μM) limited clinical application [31]. Conversely, HF exhibited poor efficacy against influenza A/B, respiratory syncytial virus, and herpesviruses. Moreover, compared with the anti-HBV agent 2′,3′-dideoxycytidine or robustaflavone, which possesses HBV-suppressive activity, HF showed no inhibitory effect on HBV (EC_50_ > 100 μM) [111,112,113]. This selectivity is instructive; it suggests that HF’s chemopreventive potential via anti-viral means may be specific to certain oncogenic viruses like EBV and HIV, but not others like HBV and likely HPV (though untested). In silico studies suggest HF may inhibit SARS-CoV-2. Mondal et al. demonstrated stronger binding interactions between HF and the SARS-CoV-2 S2 protein heptad repeat regions compared to the positive control drug nefamostat, potentially blocking viral-host membrane fusion [114]. Sawant et al. and Belhassan et al. confirmed via molecular docking and dynamics that HF and related phytochemicals bind SARS-CoV-2 main protease (Mpro) with binding affinities exceeding that of remdesivir [115,116], indicating potential as COVID-19 therapeutics. Collectively, the broad-spectrum yet selective antiviral activity of HF contributes to its anticancer profile in two significant ways: (i) by directly targeting oncogenic viruses to prevent virus-driven carcinogenesis, and (ii) by eliminating pathogenic bacteria that create a chronic inflammatory state. Since chronic inflammation is a recognized hallmark of cancer that fuels tumor progression and immunosuppression within the TME [105,106], HF’s antimicrobial and anti-inflammatory actions synergize with its direct cytotoxic mechanisms to comprehensively suppress tumorigenesis.

Beyond antiviral activity, HF shows significant antibacterial effects. Negm et al. isolated purified HF and structural derivatives from *Cycas thouarsii*, identifying potent antibacterial activity specifically attributable to purified HF—rather than extract mixtures—against *K. pneumoniae* clinical isolates, with minimum inhibitory concentrations of 0.25–0.5 μg/mL [117]. HF also exerts activity against methicillin-resistant *Staphylococcus aureus* (MRSA) by inhibiting caseinolytic protease P (ClpP; IC_50_ = 34.36 μg/mL), a virulence factor regulating toxin production and biofilm formation. In a murine MRSA-induced lethal pneumonia model, HF combined with vancomycin significantly improved survival rates (from 60% to 70% compared to vancomycin alone) and reduced lung bacterial burden, demonstrating in vivo validation of ClpP-mediated virulence attenuation and synergistic potential with standard antibiotics [118]. Computational studies by Aruwa et al. corroborate HF’s antimicrobial mechanism, with molecular docking and molecular dynamics simulations predicting that HF and robustaflavone also inhibit accessory gene regulator A (AgrA), a quorum-sensing transcription factor controlling MRSA virulence gene expression [49]. These findings suggest that HF could target multiple virulence pathways in MRSA (ClpP and AgrA), though experimental validation of AgrA inhibition remains pending.

HF demonstrates anti-inflammatory activity validated in both in vivo and ex vivo systems. Lale et al. reported that HF significantly inhibits endotoxin- and IL-1β-induced cytokine expression in human monocytes, with IC_50_ values of 18 ± 3 nM and 48 ± 4 nM, respectively [119]. Shim et al. isolated HF and 7′-O-methyl HF from *Selaginella tamariscina* and demonstrated concentration-dependent suppression of lipopolysaccharide (LPS)-stimulated NF-κB, ERK1/2, iNOS, and COX-2 expression in RAW 264.7 macrophages and HT-29 colon epithelial cells via Western blot and ELISA, resulting in decreased production of NO, IL-6, IL-8, and TNF-α [61]. In vivo evidence supporting HF’s anti-inflammatory potential derives from studies of HF-rich plant extracts. While *Juniperus rigida* fruit extracts containing HF as a major constituent alleviated oxazolone- and 2,4-dinitrochlorobenzene-induced atopic dermatitis in murine models—reducing epidermal thickness, inflammatory cell infiltration, and serum IgE levels. These findings support HF as a key contributor to the anti-inflammatory effects, though bioassay-guided fractionation is needed to confirm HF as the principal active component [52]. Mechanistically, El-Banna et al. employed network pharmacology and molecular docking to predict HF’s anti-inflammatory targets, followed by experimental validation in ex vivo human blood samples, a physiologically relevant model bridging in vitro cell lines and in vivo organismal responses. Specifically, HF treatment of LPS-stimulated peripheral blood leukocytes isolated from healthy donors significantly downregulated the expression of pro-inflammatory cytokines, including TNF-α, IL-6 and IL-1β [120]. This ex vivo approach is particularly valuable because it utilizes primary human immune cells in their native heterogeneity rather than clonal cell lines, better recapitulating in vivo immune responses and enhance translational relevance.

Collectively, HF exhibits broad and selective antiviral and antibacterial activities, effectively targeting viruses including EBV and HIV and bacteria such as *K. pneumoniae* and MRSA, but shows limited activity against HBV. Its anti-inflammatory action involves downregulation of key signaling pathways, including ERK1/2, iNOS, and COX-2, and suppression of pro-inflammatory cytokines TNF-α and IL-6. Notably, HF’s suppression of NF-κB signaling—a master regulator linking chronic inflammation to cancer—also plays a critical role in its direct antitumor cytotoxicity by removing survival signals that counteract apoptosis (detailed mechanisms are elaborated in Section 3.1). These findings suggest HF’s potential in tumor prevention through pathogen clearance and inflammation attenuation, though its inhibitory effects on crucial oncogenic pathogens like HPV and *Helicobacter pylori* remain unexplored and warrant future investigation.

In summary, HF reshapes the TME and inhibits tumor progression directly through multi-target regulation of oncogenic signaling pathways, highlighting its potential as a broad-spectrum anticancer agent. These mechanisms and related pharmacological effects across different cancer models are systematically summarized in Table 2. The multifaceted mechanisms by which HF directly targets tumor cells and remodels the TME underscore its potential as a broad-spectrum anticancer agent. However, for any potential therapeutic, a favorable efficacy profile must be coupled with an acceptable safety and toxicological profile.

### 3.6. Safety, Toxicity, and Pharmacological Limitations

While the preceding sections have detailed the promising anticancer activities of HF, a critical evaluation of its safety profile, potential toxicities, and inherent limitations is imperative to assess its true translational potential. The available data, though preliminary, offer valuable insights. A key advantage of HF is its apparent selective cytotoxicity towards cancer cells over normal cells. As summarized in Table 1, HF exhibits significantly higher IC_50_ values (indicating lower potency) in various normal cell lines, including human fibroblasts (Bj-FB, IC_50_ > 50 μM), human hepatocytes (L02, IC_50_ = 75–159.1 μM), and monkey kidney cells (Vero, IC_50_ = 45 μM), compared to many cancer cell lines (e.g., AML-2, IC_50_ = 4.93 μM). This selectivity may be partially explained by the differential stress signaling and survival pathway addiction in malignant cells, whereby HF preferentially engages pro-apoptotic networks such as ROS–JNK–p38, mitochondrial apoptosis, and NF-κB suppression in highly proliferative, oncogene-driven tumor cells, whereas quiescent or mildly stressed normal cells exhibit higher tolerance and require higher HF concentrations to undergo cytotoxic responses.

Furthermore, several studies highlight HF’s hepatoprotective effects against chemical-induced injury, a notable finding given the liver’s vulnerability to drug toxicity. HF was shown to ameliorate liver damage in models of CCl_4_-induced and APAP-induced hepatotoxicity, in some cases with efficacy comparable to the standard drug silymarin [100,102]. This suggests a favorable safety profile in the liver, a critical organ for drug metabolism. However, HF’s pharmacological profile also presents notable challenges and limitations. Its multitarget nature, while advantageous for efficacy, raises the possibility of off-target effects that have not yet been fully characterized. The most intriguing limitation is the context-dependent duality of its action on key signaling pathways. For instance, HF can inhibit the pro-survival AKT/STAT3 signaling in cancer cells to promote death [72], yet it activates the SIX4/AKT/STAT3 axis to protect hepatocytes from injury [102]. This paradox underscores that HF’s effects are highly dependent on the cellular and pathological context, which could complicate its therapeutic application. Currently, critical data on chronic toxicity, genotoxicity, and dose-limiting toxicities in vivo are absent. Future investigations must prioritize comprehensive toxicological studies in relevant animal models to define the therapeutic window, identify potential target-organ toxicities outside the liver, and establish safe dosing regimens before any clinical advancement can be considered.

## 4. Pharmacokinetic Properties and Formulation Development of HF

The transition of HF from bench to bedside depends not only on its pharmacological effects but also on its PK profile and deliverability. This section therefore reviews the two main aspects: the current understanding of HF’s PK and the formulation strategies reported to enhance its performance.

HF exhibits poor aqueous solubility and limited oral absorption, resulting in insufficient systemic exposure. The low oral bioavailability of HF results from a sequential cascade of barriers rather than a single factor. It originates from its intrinsically poor aqueous solubility, which restricts dissolution and initial absorption in the gastrointestinal tract. This is compounded by its extensive and rapid pre-systemic metabolism, which significantly depletes the parent compound before it reaches systemic circulation. The following PK data and metabolic studies quantitatively substantiate these limitations, while the advanced formulation strategies discussed thereafter are rationally designed to address these specific hurdles in a targeted manner.

In 2017, Yin et al. established the first bioanalytical method for HF and characterized its PK profile. The team developed and validated a highly sensitive liquid chromatography-tandem mass spectrometry assay with a lower limit of quantification in rat plasma reaching 0.9 ng/mL. Subsequently, they investigated the PK parameters following a single intravenous administration of 1.0 mg/kg HF in male Wistar rats. The elimination kinetics of HF displayed a biexponential decay with a half-life of 6.10 ± 1.86 h. The area under the plasma concentration-time curve from time zero to infinity (AUC_0−∞_) was 2541.93 ± 529.85 h·ng/mL, the apparent volume of distribution was 3.54 ± 1.54 L/kg, and clearance was 0.41 ± 0.08 L/h/kg [121]. Shan et al. developed an ultra-fast liquid chromatography-tandem mass spectrometry method to simultaneously quantify four flavonoid compounds, including HF, in rat plasma and applied this technique to study the PK of *Platycladus orientalis* leaf extract. Following oral administration of 3.2 g/kg of the extract, HF exhibited rapid absorption with a time to maximum concentration (T_max_) of approximately 1.9 h, a maximum plasma concentration (C_max_) of 138.45 ± 12.33 ng/mL, an elimination half-life of 2.11 ± 0.29 h, and an AUC_0−∞_ of 667.48 ± 94.59 μg·h/L. Although results across studies vary due to differences in analytical instrumentation, drug batches, and rat strains, a comparison of oral apparent clearance versus intravenous clearance consistently indicates that HF possesses very low oral bioavailability [122] (Table 3).

Chen et al. systematically characterized the metabolic profile of HF using Ultra Performance Liquid Chromatography–Quadrupole Time-of-Flight Tandem Mass Spectrometry both in vivo and in vitro, identifying 41 and 49 metabolites, respectively. They proposed that phase I metabolism primarily involves cleavage of the C-O-C ether bond yielding monomeric flavonoids, whereas phase II metabolism mainly entails conjugations such as glutamine and glycine coupling. Comparative analysis revealed a greater abundance of metabolites in the gut microbiota than in hepatic microsomes. Among in vivo biological samples, the most metabolites were detected in feces, whereas fewer were found in plasma, bile, and urine, reaffirming HF’s low systemic bioavailability [123]. This extensive metabolic clearance, particularly the cleavage of the critical C-O-C ether bond, not only limits the plasma concentration of the intact, pharmacologically active HF dimer but also raises pertinent questions about the contribution of its monomeric metabolites to the observed in vivo effects.

Biflavonoids as natural products face critical developmental challenges stemming from their poor water solubility and low oral bioavailability, impeding achievement of therapeutically effective in vivo concentrations and severely limiting clinical translation [124]. To overcome these limitations, formulation strategies aiming to enhance solubility, dissolution rate, and oral absorption have been actively pursued, demonstrating improved PK profiles and efficacy in vitro and in vivo (Figure 4).

Chen et al. employed solid dispersion technology to develop a novel oral delivery system—a polyvinylpyrrolidone K-30-based total biflavonoid extract from *Selaginella doederleinii* amorphous solid dispersion (TBESD-ASD). This formulation significantly enhanced the solubility and dissolution rates of the principal biflavonoids. In animal models, TBESD-ASD exhibited superior oral bioavailability and antitumor efficacy compared to free TBESD in xenograft-bearing mice [125]. This approach primarily tackles the initial solubility and dissolution rate barrier, facilitating the crucial first step of absorption. Although PK parameters for HF were not individually assessed in this study, HF is one of the primary biflavonoids present in *S. doederleinii* [62], suggesting that the ASD formulation may have contributed to improved PK characteristics of HF, thereby potentially enhancing its synergistic antitumor activity alongside other biflavonoids.

Polymer/surfactant self-assembled systems have also been constructed to enhance biflavonoid systemic exposure. A mixed nanomicellar system comprising D-α-tocopheryl polyethylene glycol succinate (TPGS) and polyvinyl caprolactam-polyvinyl acetate-polyethylene glycol graft copolymer (Soluplus^®^) has emerged as a promising platform for enhancing the solubility of biflavonoids [128]. Building on the demonstrated success of this platform in improving the bioavailability of related biflavonoids such as AF [129], Chen et al. applied this strategy to HF. Similarly, they encapsulated HF with TPGS and Soluplus^®^, incorporating the mitochondrial-targeting agent DQA, to prepare HF-loaded composite micelles with 91.56% encapsulation and favorable stability. Both in vitro and in vivo results demonstrated significantly enhanced cytotoxicity against A549 cells and tumor volume reduction in nude mice, without appreciable systemic toxicity, suggesting improved antitumor efficacy and mitochondrial targeting capability of HF nanoformulations for lung adenocarcinoma therapy [126]. Nanomicellar systems offer a dual advantage: they greatly enhance solubility through encapsulation and can be functionally modified to alter tissue distribution and subcellular targeting, potentially shielding the payload from premature metabolism.

Peng et al. developed a zeolitic imidazolate framework-8 (ZIF-8)-based delivery system encapsulating HF, modified with polyethylene glycol (PEG) to optimize particle size and dispersion, yielding a PEG/ZIF-8@HF complex with 92.12% encapsulation efficiency for oral melanoma treatment. PEGylation enhanced plasma stability, and acid-triggered selective release was achieved within TMEs. In vitro assays showed effective inhibition of B16F10 melanoma cell migration and invasion. In vivo, PEG/ZIF-8@HF demonstrated robust antitumor activity and induced stronger ROS-mediated mitochondrial apoptosis than free HF, positioning this ZIF-8-based oral formulation as a novel therapeutic strategy for melanoma [127]. The PEG/ZIF-8@HF system represents a more sophisticated, stimuli-responsive strategy. It not only improves solubility and stability but also enables pH-triggered release within the acidic TME, thereby maximizing local drug concentration and minimizing systemic metabolic degradation. A comparative summary of the pharmacological activity enhancements achieved by different HF formulations relative to free HF is provided in Table 4.

## 5. Discussion and Perspectives

### 5.1. Unified Hierarchical Model of HF’s Anticancer Mechanisms and Context Dependence

While the preceding sections detailed HF’s diverse anticancer activities, these mechanisms are interconnected within a hierarchical regulatory network. We propose a unified model integrating HF’s molecular targets into three functional tiers. At the apex, designated as Tier 1, lies the p53-MDM2 axis, which serves as the central molecular switch controlling cell fate decisions. HF directly inhibits MDM2, thereby stabilizing p53, which subsequently orchestrates downstream pro-apoptotic gene transcription, including Bax and Puma, enforces cell cycle checkpoints via p21 upregulation, and initiates metabolic reprogramming [22]. This p53 activation is most pronounced in wild-type p53 tumors, thereby providing a molecular rationale for patient stratification in future clinical development. At Tier 2, HF modulates several core signaling hubs that amplify and diversify its anticancer effects. First, HF activates the ROS/JNK/p38 MAPK stress-response axis, wherein mitochondrial ROS generation triggers stress-responsive kinases that amplify p53-mediated apoptosis and concurrently suppress pro-survival NF-κB signaling [64,84]. Notably, this pathway exhibits context-dependent regulation: HF functions as an antioxidant in normal hepatocytes yet acts as a pro-oxidant in tumor cells, high-lighting redox-dependent selectivity [64,65,102]. Second, HF achieves NF-κB suppression by inhibiting IκBα phosphorylation and preventing p65 nuclear translocation, thereby removing anti-apoptotic signals, and reducing inflammatory cytokine production [64]. Third, in specific therapeutic contexts such as cisplatin-resistant bladder cancer, HF inhibits casein kinase 2, which normally phosphorylates oncogenic substrates including AKT, STAT3, and NF-κB, thus representing an alternative mechanistic entry point for p53-deficient malignancies [72].

At Tier 3, these upstream regulatory signals converge on distinct phenotypic outcomes. Apoptosis execution proceeds through Bax/Bcl-2 imbalance and subsequent caspase cascade activation. This mechanism has been validated in vivo in MDA-MB-231 breast cancer [83], SMMC-7721 HCC [64], and HCT116 colorectal cancer xenograft models [84]. Cell cycle arrest is mediated through p21-dependent CDK inhibition, manifesting in con-text-dependent checkpoint selection: G0/G1 arrest predominates in HCC and colon cancer models [64,84], whereas G2/M arrest characterizes leukemia cells and S phase arrest occurs in melanoma [63,65,68]. Anti-metastatic effects emerge through MMP2 and MMP9 downregulation coupled with EMT reversal, as validated by immunohistochemical analyses in xenograft tumor tissues [65,84]. A critical feature of this unified model is context-dependent pathway utilization, wherein HF’s molecular effects vary according to cellular p53 status, tumor type, redox microenvironment, and stromal interactions. This hierarchical framework provides a mechanistic foundation for rational clinical development strategies, including biomarker-driven patient selection focused on wild-type p53 and MDM2-overexpressing tumors, combination therapeutic approaches pairing HF with DNA-damaging chemotherapeutics or immune checkpoint inhibitors, and mechanism-based dose optimization tailored to specific molecular targets. Collectively, this integrated model unifies HF’s anticancer mechanisms into a cohesive hierarchical network wherein p53 stabilization serves as the central regulatory node, amplified by parallel ROS/JNK and NF-κB signaling hubs, ultimately converging on apoptosis induction, proliferation arrest, and metastasis suppression with inherent tumor-selective cytotoxicity.

### 5.2. Comparative Analysis of HF with Structurally Related Amentoflavone and Clinical MDM2 Inhibitors

While HF belongs to the C-O-C-type biflavonoid class, AF represents the C-C-type subclass, with their structural difference mainly arising from the position of interflavone linkage [22,130]. This structural diversity results in distinct pharmacological properties. AF consistently displays superior in vitro antiproliferative potency across multiple cancer cell lines, with reported IC_50_ values typically ranging from 1 to 30 μM [131,132,133], whereas HF’s IC_50_ values often fall within 10 to 80 μM in similar models [63,64,84]. Notably, the antitumor efficacy of AF has been validated in vivo. In a HCC xenograft model, oral administration of AF at 100 mg/kg per day resulted in significant tumor growth inhibition [134]. Similarly, in preclinical models of pancreatic cancer, daily oral dosing at 80 mg/kg significantly suppressed tumor growth and exerted marked pro-apoptotic and anti-migratory effects [135]. Both compounds activate p53, inhibit NF-κB, and reverse EMT, but AF and HF show distinct target preferences at the molecular level [35,136]. AF has been shown to suppress tumorsphere formation by regulating the Hedgehog/Gli1 signaling pathway in breast cancer stem cells [137], and also uniquely triggers ferroptosis through regulation of the AMPK/mTOR pathway [138], mechanisms which have not been characterized for HF. In contrast, HF has two distinctive advantages: strong inhibition of CK2 with an IC_50_ of about 1 μM, which enables reversal of cisplatin resistance in bladder cancer cells [72], and modulation of pre-mRNA splicing through blocking spliceosome progression [69], a novel mechanism that has not been reported for AF. Both compounds suffer from poor oral bioavailability, with AF reaching approximately 8–15% in rats due to extensive hepatic first-pass metabolism [139], and HF displaying similarly low bioavailability in PK studies [121]. The C-C bond in AF also provides greater metabolic stability compared with the more labile C-O-C bond in HF [130]. Despite these differences, both biflavonoids remain at the early preclinical stage with no clinical trials initiated [22,121], and both face similar translational barriers such as lack of formal toxicology studies or patient-derived xenograft validation.

HF functions as a potent MDM2 inhibitor and thus shares the same therapeutic target as clinical-stage MDM2 antagonists. However, important mechanistic and pharmacological differences warrant careful examination. Placing HF in the context of clinical-stage synthetic MDM2 inhibitors—most notably the Nutlins (e.g., RG7112) and idasanutlin (RG7388), which are in advanced phase trials—reveals both opportunities and limitations [82]. Synthetic MDM2 antagonists are characterized by high affinity (nanomolar) for MDM2 and robust efficacy in p53 wild-type tumors, but they often induce dose-limiting hematological toxicities like thrombocytopenia, due to on-target p53 activation in the bone marrow [140]. HF exhibits a multi-target profile, acting on p53/MDM2 [63], ROS/JNK [64,84], NF-κB [64,84], and CK2 [72], which could offer reduced potential for single-target resistance, lower therapeutic doses via pathway convergence, and possibly an improved toxicity profile, as suggested by the absence of overt adverse effects in available xenograft studies [64,83,84]. However, its lower potency in vitro compared to these clinical candidates and the lack of human safety data remain significant hurdles. The clinical progress of Nutlins and idasanutlin validates the p53/MDM2 pathway as a druggable target in oncology [82], underlining a potential niche for HF especially in p53 wild-type, MDM2-amplified cancers. The key translational question is whether HF’s broader mechanism, trading single-target potency for multi-pathway activity, can achieve therapeutic windows superior to or complementary with established MDM2 inhibitors, a hypothesis testable only through rigorous head-to-head preclinical comparisons and eventual clinical trials.

A strategic approach for HF development should focus on differentiating it from AF by emphasizing its unique CK2 inhibition [72] and splicing modulation activity [69], targeting resistant tumors and mechanistically distinct cancer types. Combined therapy with existing MDM2 inhibitors is rational, as HF-induced ROS generation and NF-κB inhibition [64,84] may synergize with these agents to counteract resistance. Formulation improvements, such as use of TPGS/Soluplus micelles and metal–organic frameworks [127], are expected to improve HF’s bioavailability and represent an advantage over AF, which lacks such progress. Finally, patient selection based on p53 status, MDM2, and CK2 expression [63,72], as suggested by comparative mechanistic studies, may expedite trial success, as has been demonstrated for targeted therapies [141,142].

### 5.3. Critical Evaluation of Inconsistent Findings and Limitations in HF Research

Despite substantial evidence confirming HF as a promising antitumor lead compound exhibiting multifaceted pharmacological activities, several contradictory findings and limitations remain, reflecting gaps in the current evidence base. These controversial results and opposing evidence from different studies are summarized in Table 5.

Firstly, evaluation of HF’s antioxidant capacity and hepatoprotective effects exhibits notable discrepancies. Utilizing DPPH-UPLC-Q-TOF/MS analysis, Wang et al. identified HF as the most potent radical scavenger among several purified biflavonoids, including podocarpusflavone A, bilobetin, and ginkgetin [62]. In contrast, assessments of *Selaginella sinensis* extracts via HPLC-DPPH demonstrated that purified HF exhibited only appreciable DPPH scavenging activity at a relatively high concentration of 75 μM, substantially weaker than quercetin (IC_50_ = 3.2 ± 0.02 μM) and rutin (IC_50_ = 3.8 ± 0.03 μM) [98]. These variances may be attributed to differences in extract composition, plant source heterogeneity, uncertainties in HF purity, and methodological factors, or HF purity, leading to inconsistent antioxidant potency estimations. A similar pattern of inconsistency is observed for hepatoprotection. Evidence supports HF’s hepatoprotective effects in CCl_4_-induced liver injury models, showing comparable efficacy to standard silymarin treatment and amelioration of hepatic histopathology [100]. Nevertheless, a separate study revealed that HF alone or combined with glycyrrhizin, while superior to either agent alone, failed to outperform silymarin in protective efficacy [101]. Additionally, Liu et al. demonstrated HF’s hepatoprotection in APAP-induced injury models via SIX4/AKT/STAT3 pathway activation, mitigating oxidative stress, inflammasome activation, and pyroptosis [102]. These conflicting results indicate that HF’s hepatoprotective mechanisms—antioxidative and signaling pathways—may possess limited applicability across varying hepatic injury contexts. Considering the common progression from hepatic injury to HCC, such inconsistencies may diminish HF’s cancer preventive potential, highlighting the urgent need for standardized, high-quality in vivo studies clarifying the relationship between HF’s hepatoprotection and hepatocarcinogenesis prevention.

HF’s reported antimicrobial and antimetastatic activities show pronounced selectivity across different pathogens and tumor models, and are still supported mainly by limited in vivo evidence. HF’s antimicrobial profile is highly selective with in vitro and in silico studies confirming potent inhibition of dengue virus NS5 RdRp and NS5 protein [25], EBV-EA activation [110], HIV-1 reverse transcriptase [31], and SARS-CoV-2 Mpro [115,116] and S2 fusion protein [114]. HF also exhibits considerable activity against MRSA [118]. However, HF lacks efficacy against influenza A/B viruses, respiratory syncytial virus, herpesviruses [111,112], and HBV [113]. Moreover, conflicting antibacterial results against *Klebsiella pneumoniae* have been reported [117,143], coupled with HF’s high cytotoxicity, these features currently limit its direct translational potential in anti-infective settings [31]. In this context, and considering the role of pathogenic infections in oncogenesis [107], HF’s pathogen-specific activity may provide novel avenues for cancer prevention, yet evaluation of HF’s in vivo antimicrobial efficacy with respect to tumor prevention remains necessary. Similarly, studies of HF’s antimetastatic effects are notably limited in vivo. Although in vitro evidence demonstrates suppression of MMP2/9 expression, upregulation of TIMP2 [84], inhibition of EMT markers [83], and diminished migratory/invasive capacity [65], in vivo xenograft models mostly use immunohistochemical detection of MMP9 [84] and MMP2 [83] to infer antimetastatic potential. The lack of direct functional in vivo validation, particularly in orthotopic or metastatic models integrating imaging modalities, risks overestimation of HF’s clinical efficacy against metastasis. Future research should address these gaps to substantiate the translational value of HF’s antimetastatic activity.

The promising pharmacological profile of HF is tempered by notable inconsistencies in reported data, which demand a critical look beyond their simple cataloging. A primary source of variability lies in the inherent phytochemical complexity of its numerous plant sources; the bioactivity of an HF-containing extract from *Selaginella doederleinii*, where it was the dominant antioxidant [62], cannot be directly equated to one from *Selaginella sinensis*, where its effect was masked by stronger antioxidants like quercetin [98]. This “matrix effect” is compounded by significant methodological divergences across studies. For instance, divergent results on its hepatoprotective efficacy [100,101] may reflect the use of distinct injury models (CCl_4_ vs. APAP) that engage different pathological mechanisms, a nuance that is often overlooked when comparing outcomes. The reliance on different in vitro assays, from chemical radical-scavenging tests to cellular models, further complicates cross-study comparisons, as does the use of varying bacterial strains in antimicrobial tests [117,143]. Underpinning these issues is the critical uncertainty regarding the purity of the HF tested in many studies and its extensive in vivo metabolism, which can cleave the characteristic C-O-C bond and generate active monomers and conjugates [123], meaning the administered compound may not be the active species. Therefore, these inconsistencies do not necessarily invalidate HF’s potential but rather highlight its context-dependent actions and underscore the non-trivial challenge of reproducing results without standardized, well-characterized materials and mechanistic-driven study designs.

Limited and fragmented PK information further constrains the interpretation of these findings and HF’s translational prospects. Existing pharmacokinetic data are largely restricted to Yin et al.’s intravenous rat model [121] and Shan et al.’s oral administration study [122]. Moreover, the significant role of the gut microbiota in HF metabolism presents a critical research gap that extends beyond just PK. The finding that gut microbiota are highly active in metabolizing HF in vitro raises an important question regarding its in vivo action [123]. It remains unknown whether the resulting metabolites, such as the apigenin monomers from ether bond cleavage, are pharmacologically active, leaving their contribution to overall antitumor efficacy and systemic effects unresolved. Beyond these fundamental PK parameters, data are also critically lacking on drug–drug and food-drug interactions. To our knowledge, no studies have yet investigated the impact of food matrices on the oral absorption of HF, which is a crucial factor given its already low bioavailability. Meanwhile, HF’s potential to inhibit or induce key drug-metabolizing enzymes has not been characterized. Collectively, these gaps in pharmacokinetics, metabolism, and interaction profiles represent key limitations that must be addressed before HF can be safely evaluated in clinical trials.

### 5.4. Clinical Translation Strategies and Future Perspectives

To effectively translate HF’s promising preclinical findings into clinical benefit, future development should focus on three strategic clinical positionings: (1) Given that HF’s cytotoxicity is notably higher in p53-wild-type cells compared to p53-null or mutant lines, future clinical trial designs should consider p53 status as a predictive biomarker. Stratifying patients who retain wild-type p53 could maximize HF’s therapeutic response and clinical success rate. (2) HF’s ability to reverse cisplatin resistance and target the SENP1 suggests it acts not merely as a monotherapy, but as a potent chemosensitizer. Clinically, HF could be positioned in combination regimens for patients with refractory or multidrug-resistant tumors, aiming to restore sensitivity to standard-of-care chemotherapeutics. (3) The successful development of bioavailability-enhanced oral formulations is a critical step towards clinical utility. Unlike intravenous agents, these oral delivery systems enable HF to be developed for chronic administration, potentially serving as a maintenance therapy to prevent metastasis or recurrence in the post-operative setting.

Furthermore, successful clinical translation of HF requires systematic and rigorous resolution of several critical knowledge and development gaps. First, in-depth PK and toxicological profiling is imperative. Beyond improving oral bioavailability, future studies must define the therapeutic window, identify potential target-organ toxicities beyond the liver, and characterize the safety profile of HF’s major metabolites. Second, the precise molecular mechanisms require further elucidation. The context-dependent duality of HF, such as its opposing roles in the AKT/STAT3 pathway in cancer cells versus stressed hepatocytes, demands a systems-level understanding using multi-omics approaches and genetically engineered animal models to identify predictive biomarkers for patient stratification. Third, the battle against tumor heterogeneity and resistance must be proactive. Rational combination therapies—pairing HF with standard chemotherapeutics, targeted agents, or immunotherapies—should be explored to exploit potential synergies and overcome mechanisms like p53-dependent cytotoxicity. Finally, the full potential of HF will only be realized through smarter delivery. The next generation of formulations should evolve from simple solubility enhancers to intelligent, tumor-microenvironment-responsive systems capable of co-delivering synergistic drugs and targeting specific cell populations within the tumor. By systematically tackling these translational barriers—safety, mechanism, combination strategies, and advanced delivery—HF can truly transition from a pharmacological tool to a novel clinical modality for cancer treatment.

In summary, this review highlights HF as a structurally unique C–O–C-linked biflavonoid with multitarget anticancer activity, integrating p53/MDM2 stabilization, Bcl-2/Bax modulation, regulation of ROS/JNK and NF-κB signaling, cell cycle arrest, and inhibition of metastasis. Although poor solubility, low and variable oral bioavailability, and incomplete pharmacokinetic and safety data currently limit immediate clinical application, recent progress in nanodelivery systems and mechanistic understanding, together with opportunities for biomarker-guided patient selection and rational combination therapy, supports an optimistic view that HF may ultimately evolve into a clinically useful option for selected patients with treatment-refractory, p53 wild-type malignancies.

## Figures and Tables

**Figure 1 cells-15-00017-f001:**
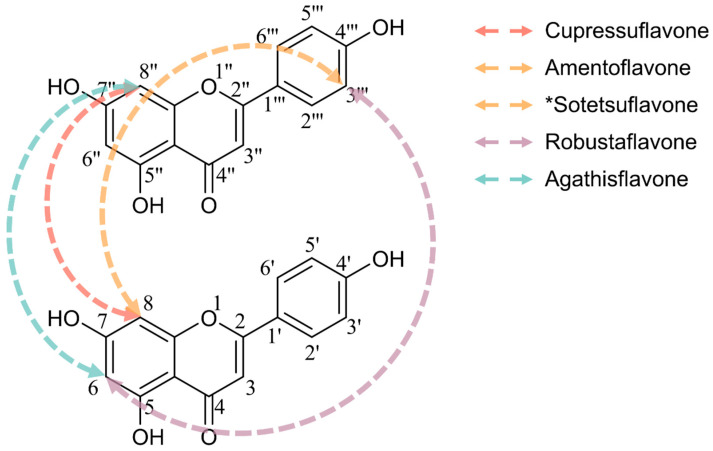
Structural diversity of C-C linked biflavonoids. * AF and sotetsuflavone are both linked between the 8- and 3‴-positions. Sotetsuflavone is distinguished by a methoxy group (-OCH_3_) at the 7-position.

**Figure 2 cells-15-00017-f002:**
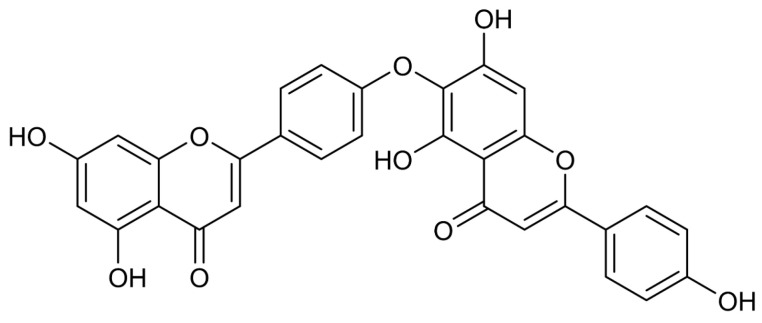
Chemical structure of HF. HF is a typical C-O-C type bioflavonoid, consisting of two apigenin units linked through the 4′- and 6″-positions by an ether bond.

**Figure 3 cells-15-00017-f003:**
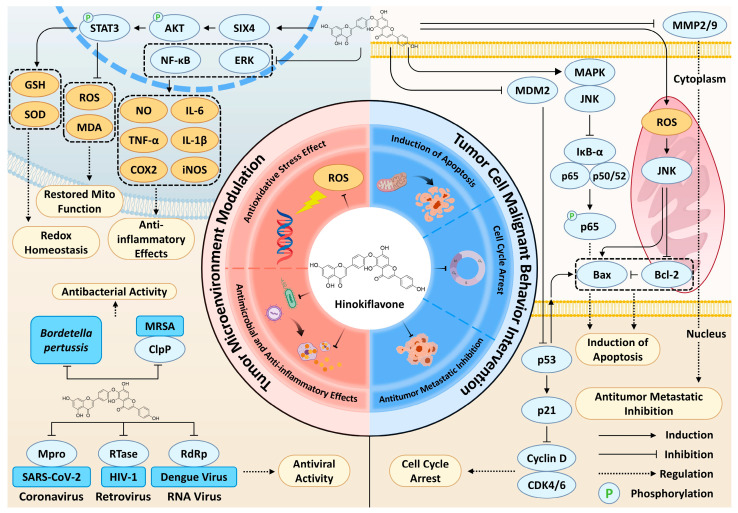
An overview of the dual-action anticancer mechanisms of HF. HF exhibits its therapeutic potential by acting on two interconnected fronts: modulating the TME and directly intervening in tumor cell malignant behavior. On one hand, HF directly targets tumor cells to induce apoptosis by activating the p53 and ROS-JNK signaling axis, suppressing NF-κB through the inhibition of p65 phosphorylation, which collectively disrupt the Bax/Bcl-2 balance. The stabilization of p53 also triggers cell cycle arrest via the p21-CDK4/6 axis. Furthermore, HF impedes tumor metastasis by downregulating key matrix metalloproteinases, including MMP2 and MMP9. On the other hand, HF reshapes the TME through its antioxidative, anti-inflammatory, and antimicrobial/antiviral properties. It sup-presses pro-inflammatory pathways such as NF-κB, ERK, thereby reducing inflammatory factors (e.g., TNF-α, IL-6, etc.). HF alleviates oxidative stress and maintains redox homeostasis, in part via modulation of the SIX4/AKT/STAT3 axis (context-dependent). In addition, HF displays antibacterial activity against MRSA and Bordetella pertussis and exhibits broad-spectrum antiviral activity by targeting viral enzymes (Mpro, RTase, RdRp). This multifaceted mechanism highlights HF’s capacity to synergistically disrupt both the supportive niche and the intrinsic malignancy of cancer cells.

**Figure 4 cells-15-00017-f004:**
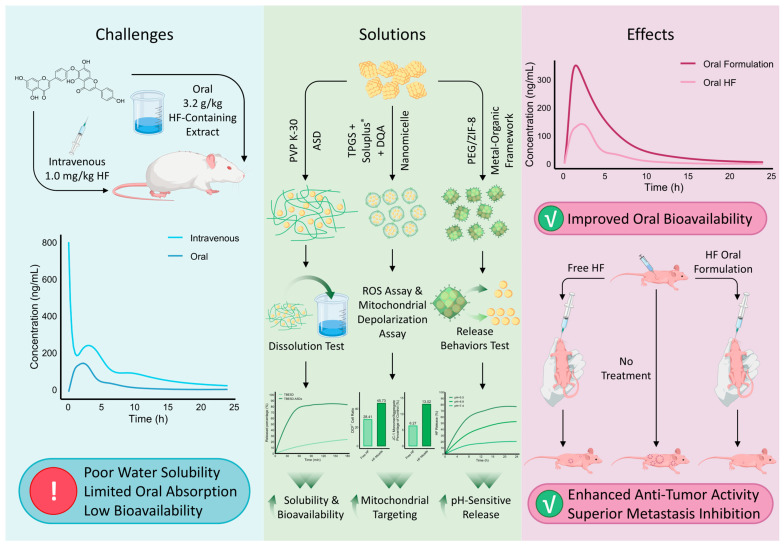
Challenges, formulation strategies, and therapeutic benefits of HF oral delivery. HF suffers from poor aqueous solubility, limited oral absorption, and low bioavailability, as illustrated by the marked differences in plasma concentration-time curves between intravenous and oral administration. To overcome these challenges, diverse formulation strategies have been developed, including ASD with PVP K-30 [125], nanomicelles (TPGS/Soluplus^®^ combined with the mitochondrial-targeting ligand dequalinium chloride (DQA)) [126], and metal–organic frameworks (PEG/ZIF-8) [127]. These approaches enhance dissolution, promote mitochondrial targeting, and enable pH-sensitive release, as verified by dissolution profiles, ROS assays, mitochondrial depolarization assays (The bar chart values were sourced from Chen et al. [126]), and release behavior tests. Compared with free HF, optimized oral formulations significantly improve bioavailability, potentiate antitumor efficacy, and confer superior inhibition of metastasis in vivo. Together, these findings underscore the potential of advanced formulation strategies to translate HF into an effective oral anticancer agent.

**Table 1 cells-15-00017-t001:** Summary of reported cytotoxicity (IC_50_) of HF against different cell lines.

Cell Type	Cell Line	Time Point (h)	IC_50_ (μM)	Assay Method	Refs
Leukemia	AML-2	24	4.93 ± 1.16	Celltiter Glo	[63]
Leukemia	HL-60	24	10.95 ± 0.19	Celltiter Glo	[63]
Chronic Myeloid Leukemia	K562	24	23.38 ± 1.78	CCK-8 Assay	[68]
Chronic Myeloid Leukemia	K562	48	8.84 ± 1.62	CCK-8 Assay	[68]
Colorectal Cancer	HCT116 p53-deficient	24	32.66 ± 0.31	Celltiter Glo	[63]
Colorectal Cancer	HCT116	24	14.19 ± 2.04	Celltiter Glo	[63]
Colorectal Cancer	HCT116	48	13	MTT Assay	[84]
Colorectal Cancer	CT26	48	12	MTT Assay	[84]
Colorectal Cancer	HT29	48	13	MTT Assay	[84]
Colorectal Cancer	SW48	48	14	MTT Assay	[84]
Colorectal Cancer	SW480	48	17	MTT Assay	[84]
Colorectal Cancer	DLD-1	48	17	MTT Assay	[84]
Colorectal Cancer	SW620	48	18	MTT Assay	[84]
Osteosarcoma	U2OS	24	15.90 ± 2.07	Celltiter Glo	[63]
Breast Cancer	MCF-7	24	17.33 ± 1.90	Celltiter Glo	[63]
Melanoma	B16	24	20	MTT Assay	[65]
Melanoma	B16	48	10	MTT Assay	[65]
Melanoma	A375	24	23	MTT Assay	[65]
Melanoma	A375	48	10	MTT Assay	[65]
Melanoma	CHL-1	24	25	MTT Assay	[65]
Melanoma	CHL-1	48	12	MTT Assay	[65]
Breast Cancer	MDA-MB-231	48	≈20	MTT Assay	[83]
Breast Cancer	4T1	48	>80	MTT Assay	[83]
HCC	SMMC-7721	24	74.4 ± 8.1	CCK-8 Assay	[64]
HCC	SMMC-7721	48	60.3 ± 2.9	CCK-8 Assay	[64]
HCC	HepG 2	24	80.8 ± 2.6	CCK-8 Assay	[64]
HCC	HepG 2	48	57.5 ± 5.3	CCK-8 Assay	[64]
Normal Human Hepatocytes	L02	24	75	MTT Assay	[65]
Normal Human Hepatocytes	L02	24	159.1 ± 5.6	CCK-8 Assay	[64]
Normal Human Hepatocytes	L02	48	104.7 ± 4.5	CCK-8 Assay	[64]
Normal Monkey Kidney Cells	Vero	24	45	MTT Assay	[65]
Normal Monkey Kidney Cells	Vero	48	29	MTT Assay	[65]
Normal Human Fibroblast Cell Line	BJ-FB	24	>50	Celltiter Glo	[63]

**Table 2 cells-15-00017-t002:** Major antitumor mechanisms and related pharmacological effects of HF.

Pharmacological Effect	Cell Line(s)	Cancer Type/Model	In Vivo Validation	Main Targets/Pathways	Mechanism Description	Refs
Induction of Apoptosis	AML-2	Leukemia	No in vivo confirmation	MDM2-p53	HF targets the MDM2-MDMX RING domain, inhibits MDM2’s E3 ubiquitin ligase activity, reducing p53 ubiquitination and degradation	[63]
	HL-60	Leukemia				
	U2OS	Osteosarcoma				
	MCF-7	Breast Cancer				
	HCT116	Colorectal Cancer	No in vivo confirmation		HF time- and dose-dependently suppresses *MDM2* mRNA synthesis, relieving MDM2-mediated p53 inhibition	[67]
	MDA-MB-231	Breast Cancer	Mouse xenograft model (MDA-MB-231): IHC confirmation	Bax/Bcl-2	HF downregulates Bcl-2 and dose-dependently upregulates Bax, inducing caspase-dependent apoptosis	[83]
	A375	Melanoma				[65]
	CT26, HCT116	Colorectal Cancer	No in vivo confirmation			[84]
	SMMC-7721, HepG2	HCC	Mouse xenograft model (SMCC-7721): Western blot and IHC confirmation	JNK, p38	HF dose-dependently activates JNK and p38, lowering Bcl-2/Bax ratio and triggering intrinsic apoptosis	[64]
	K562	Leukemia	No in vivo confirmation		In addition to apoptosis, HF induces autophagy in K562 cells	[68]
	SMMC-7721, HepG2	HCC	Mouse xenograft model (SMCC-7721): Western blot and IHC confirmation	NF-κB	HF significantly reduces NF-κB activity by inhibiting IKBα phosphorylation and p65 nuclear translocation	[64]
	K562	Leukemia	No in vivo confirmation		In leukemia cell lines, it has been confirmed that HF inhibits NF-κB activity by activating the JNK/p38 signaling pathway	[68]
Cell Cycle Arrest	SMMC-7721, HepG2	HCC	No in vivo confirmation	CDK4, CDK6, p21	Downregulates cyclin D1, CDK4, CDK6, upregulates p53, induces G0/G1 arrest	[64]
	K562	Leukemia	No in vivo confirmation	Cdc2, p21	Upregulates p21, downregulates Cdc2, induces G2/M arrest	[68]
	HCT116	Colorectal Cancer	No in vivo confirmation	p21, 14-3-3σ	Promotes transcription of p21 and 14-3-3σ, inducing G2/M arrest	[67]
	A357, B16	Melanoma	No in vivo confirmation	-	Induces S phase arrest	[65]
Inhibition of Tumor Metastasis	CT26, HCT116	Colorectal Cancer	Mouse syngeneic model (CT-26): IHC confirmation	MMP2, MMP9, TIMP2	Inhibits MMP2 and MMP9 expression, upregulates TIMP2 expression, reducing tumor cell migration	[84]
	A375	Melanoma	No in vivo confirmation	MMP2, MMP9	HF decreases MMP2 and MMP9 levels, inhibiting tumor cell invasion and migration	[65]
	MDA-MB-231, 4T1	Breast Cancer	Mouse xenograft model (MDA-MB-231): IHC confirmation	E-cadherin, N-cadherin	Dose-dependently upregulates E-cadherin and downregulates N-cadherin, reversing or inhibiting EMT to suppress invasion	[83]
Antioxidant and Hepatoprotective Effects	-	CCl_4_-induced liver injury rats	Male Wistar rats: Histopathology confirmation	-	At biochemical level, HF’s hepatoprotective activity comparable to positive control silymarin, mechanism not detailed	[100]
	-	CCl_4_-induced liver injury in rats	Male Wistar albino rats: Histopathology, electron microscopy and enzyme activity assay	-	Combination of HF and glycyrrhizin provides less hepatoprotection versus silymarin, mechanism unclear	[101]
	-	APAP-induced drug-induced liver injury	Female C57BL/6 mice: Histopathology, Western blot and enzyme activity assay	SIX4, Akt, Stat3	HF activates SIX4-mediated Akt/Stat3 pathway, inhibiting inflammasome activation and pyroptosis induced by APAP	[102]
Anti-inflammation Effects	RAW 264.7, HT-29	-	No in vivo confirmation	ERK1/2, iNOS, COX-2	HF and mHF inhibit ERK1/2, iNOS, COX-2 expression in LPS-stimulated cells concentration-dependently, reducing NO, IL-6, IL-8, TNF-α	[61]
	Human leukocytes	-	Ex vivo human white blood cells: MTT assay and RT-qPCR	TNF-α, IL-6, IL-1β	HF inhibits expression of inflammatory cytokines TNF-α, IL-6, IL-1β; TNF-α inhibition comparable to positive control piroxicam	[120]

**Table 3 cells-15-00017-t003:** Major pharmacokinetic parameters (Mean ± SD) after single 1.0 mg/kg intravenous injection of HF and oral 3.2 g/kg *Platycladus orientalis* leaf extract (*n* = 6).

Parameter	Unit *	Oral *Platycladus orientalis* Leaf Extract (Ref. [122])	Intravenous HF (Ref. [121])
t_1/2_	h	2.11 ± 0.29	6.10 ± 1.86
AUC_0−t_	ng·h/mL	667.08 ± 94.31	2394.42 ± 466.86
AUC_0−∞_	ng·h/mL	667.48 ± 94.59	2541.93 ± 529.85
CL	L/h/kg	393.6 ± 61.8 (CL/F) **	0.41 ± 0.08 (CL)
T_max_	h	1.92 ± 0.20	-
C_max_	ng/mL	138.45 ± 12.33	-
C_2min_	ng/mL	-	803.42 ± 92.75
MRT_0−t_	h	-	6.01 ± 0.68
MRT_0−∞_	h	-	7.55 ± 1.37
V_d_	L/kg	-	3.54 ± 1.54

t_1/2_: half-life; AUC_0−t_: area under the concentration-time curve from time zero to time t; AUC_0−∞_: area under the concentration-time curve from time zero to infinity; CL: clearance rate; T_max_: time to reach maximum concentration; C_max_: maximum concentration; C_2min_: plasma concentration at 2 min; MRT_0−t_: mean residence time from zero to time t; MRT_0−∞_: mean residence time from zero to infinity; V_d_: apparent volume of distribution; * Units have been converted; ** Oral administration results represent apparent clearance (CL/F), whereas intravenous administration results represent actual clearance (CL).

**Table 4 cells-15-00017-t004:** Pharmacological activity enhancements of different HF formulations relative to free hinokiflavone.

Formulation Type	HF Hybrid Nanomicelles [127]	PEGylated ZIF-8@HF Drug Delivery System [128]
In vitro cytotoxicity	In A549 cells, HF-loaded TPGS/Soluplus + DQA micelles demonstrated a 2.48-fold increase in cytotoxic potency compared with free HF (IC_50_ = 7.81 μg/mL vs. 19.34 μg/mL).	In B16F10 melanoma cells, PEG/ZIF-8@HF exhibited an approximately 1.8-fold enhancement in cytotoxicity relative to free HF (IC_50_ ≈ 4 μM vs. 7.5 μM).
In vivo tumor inhibition	In A549 subcutaneous xenograft models, HF-micelles achieved a 1.41-fold improvement in tumor inhibition compared with free HF (tumor inhibition ratio: 64.76% vs. 45.92%).	In B16F10 melanoma–bearing nude mice, PEG/ZIF-8@HF produced an approximately 1.53-fold increase in antitumor efficacy (tumor inhibition ratio: 50.46% vs. 33.03%).
Pro-apoptotic effects	HF-micelles induced mitochondrial depolarization and apoptosis 1.57 times more effectively than free HF in vitro (47.23% vs. 30.11%).	PEG/ZIF-8@HF enhanced apoptosis induction by approximately 1.82-fold in vivo, as evidenced by a higher proportion of TUNEL-positive tumor regions (40.83% vs. 22.43%).

**Table 5 cells-15-00017-t005:** Controversial findings regarding the pharmacological effects of HF.

diPharmacological Effect	Supporting Evidence	Opposing Evidence
Antioxidant Activity	DPPH-UPLC-Q-TOF/MS evaluation showed HF had the strongest antioxidant capacity among biflavonoids from *Selaginella doederleinii* extracts [62]	HPLC-DPPH evaluation of *Selaginella sinensis* extracts showed HF only displayed DPPH scavenging activity at high concentrations, which was much weaker than quercetin and positive control rutin [98]
Hepatoprotective Effect	HF exhibited hepatoprotective effects comparable to the standard drug silymarin in CCl_4_-induced liver injury models [100]	HF alone or combined with glycyrrhizin showed better protection than either alone but did not surpass silymarin in CCl_4_-induced liver injury models [101]
Antimicrobial Activity	*Cycas thouarsii* extracts demonstrated antibacterial activity against clinical *Klebsiella pneumoniae* isolates, with HF being the most active purified component [117]	*Juniperus chinensis* L. ethanol extracts containing HF showed only weak inhibitory activity against *Klebsiella pneumoniae* [143]

## Data Availability

Data sharing is not applicable to this article as no new data were created or analyzed.

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
