# Peer review of "Hinokiflavone as a Potential Antitumor Agent: From Pharmacology to Pharmaceutics"

_cells, 2025, doi:10.3390/cells15010017_

Round 1

Reviewer 1 Report

Comments and Suggestions for Authors

The authors of this manuscript have analyzed the various biological properties of hinokiflavone, including its antitumor activity, pharmacokinetics, and bioformulation developments, highlighting challenges and prospects for clinical application.
The authors could provide some examples of natural molecules that have demonstrated significant clinical use.
The authors believe the microbiota may be involved in the action of hinokiflavone.
The authors should also indicate any interactions with food, other drugs, other natural molecules, and side effects.
The authors should place greater emphasis on potential clinical use.

Author Response

Reviewer 1, Comment 1: The authors could provide some examples of natural molecules that have demonstrated significant clinical use.

Response:

We thank the reviewer for this constructive suggestion. We agree that highlighting clinically successful natural products provides important context and strengthens the rationale of our review. In response, we have revised the Introduction section to expand upon the examples of natural product-derived drugs. In addition to the originally mentioned paclitaxel and vincristine, we have now incorporated other prominent examples, such as the camptothecin derivative irinotecan and the anthracycline doxorubicin, both of which originate from natural sources and have become cornerstones of modern chemotherapy.

(see the second paragraph of section: 1. Introduction; lines 56-61)

Reviewer 1, Comment 2: The authors believe the microbiota may be involved in the action of hinokiflavone.

Response:

We thank the reviewer for highlighting this critical and insightful point. The reviewer is correct. Our manuscript underscores this finding based on the comparative in vitro study by Chen et al. [124]. We agree that this implication was not sufficiently emphasized in our original discussion. To address this, we have now added a new discussion point in Section 5 (Discussion and Perspectives), highlighting the need for future studies to investigate the pharmacological "action" of these microbiota-generated metabolites to fully understand HF's in vivo efficacy.

(see the fifth paragraph of section 5.3. Critical Evaluation of Inconsistent Findings and Limitations in HF Research; lines 963-968)

Reviewer 1, Comment 3: The authors should also indicate any interactions with food, other drugs, other natural molecules, and side effects.

Response:

We thank the reviewer for raising this vital point, which is essential for assessing the clinical translation potential of HF. We have carefully revised the manuscript to better address these aspects.

Regarding side effects, we have added a dedicated subsection (Section 3.6. Safety, Toxicity, and Pharmacological Limitations) to detail the current safety data. Specifically, we highlight HF's favorable therapeutic window, as evidenced by significantly higher IC50 values (lower toxicity) in normal cell lines (e.g., BJ, L02, Vero) compared to cancer cell lines.

Regarding interactions with food, drugs, and other natural molecules, we performed a thorough literature search. However, current data are critically lacking. To our knowledge, no specific studies have investigated the impact of food matrices or interactions with other natural molecules on HF's absorption, nor has its potential to inhibit/induce key drug-metabolizing enzymes (e.g., CYP450) been characterized.

(see the fifth paragraph of section 5.3. Critical Evaluation of Inconsistent Findings and Limitations in HF Research; lines 968-975)

Reviewer 1, Comment 4: The authors should place greater emphasis on potential clinical use.

Response:

We appreciate this valuable suggestion. We agree that discussing the potential clinical translation is essential to elevate the review beyond preclinical pharmacology. Since HF has not yet entered clinical trials, we have strengthened the manuscript by explicitly defining its potential clinical positioning based on current mechanistic data. Based on the findings that HF's efficacy is partly p53-dependent, we propose that HF would be best positioned for patients stratified by p53 status, serving as a potential biomarker-driven therapy. We emphasized HF's ability to reverse cisplatin resistance and inhibit SENP1, suggesting its clinical utility as an adjuvant agent in refractory or metastatic cancers to re-sensitize tumors to standard chemotherapy.

We highlighted that the development of novel oral formulations paves the way for HF to be used in long-term maintenance therapy or prevention, which requires high patient compliance.

We believe these additions provide a clearer roadmap for how HF could be integrated into future clinical regimens.

(see the first paragraph of section 5.4. Clinical Translation Strategies and Future Perspectives; lines 977-989)

Reviewer 2 Report

Comments and Suggestions for Authors

This manuscript provides a broad overview of the pharmacological and anticancer activities of hinokiflavone. While the topic is timely and potentially impactful, the manuscript requires substantial revision before it can be considered for publication.

  • Throughout the manuscript, many figure citations are broken (e.g., Figure 1, 2, 3, etc.). These must be corrected so that readers can follow the structure.
  • Several sections repeat the same mechanistic pathways (ROS, NF-κB, JNK, MDM2/p53, EMT). Please reorganize and consolidate overlapping content to improve clarity and flow.
  • The manuscript jumps from pharmacodynamics to pharmacokinetics, then to antimicrobial, anti-inflammatory, and anticancer effects without clear transitions. I recommend reorganizing the review into a more cohesive narrative.
  • Many conclusions are drawn from cell-line experiments without sufficient discussion of in vivo data or clinical relevance. Please incorporate more critical evaluation of animal studies and address limitations of in vitro findings.
  • Given that HF is described as a potential therapeutic, a dedicated section discussing safety, toxicity, off-target effects, and pharmacological limitations is necessary.
  • The manuscript mentions inconsistent pharmacological results but does not critically evaluate possible causes or implications. A more rigorous comparison of conflicting reports is needed.
  • While PK parameters are described, there is little mechanistic or comparative insight. Please discuss why HF bioavailability is low, how metabolism limits therapeutic use, and how different formulations address these barriers.
  • The manuscript lists molecular mechanisms but does not integrate them into a unified model explaining HF’s anticancer effects. Please provide thematic summaries that unify the described pathways.
  • The antiviral activity is interesting but feels loosely connected to the main cancer-focused narrative. Consider tightening or more clearly linking this section to the manuscript’s main theme.
  • The review highlights HF’s benefits but provides little context relative to similar compounds (e.g., amentoflavone) or established anticancer drugs. A comparative discussion would strengthen the manuscript.
  • The manuscript describes nanomicelles, MOFs, and amorphous solids but does not provide quantitative improvements (e.g., fold-increase in solubility or bioavailability). Including comparative metrics would improve clarity.
  • Some claims (e.g., “HF demonstrates superior selective cytotoxicity”) are not sufficiently supported by data. Please moderate these claims or provide stronger evidence.
  • Figures are insufficiently referenced or explained in the text. Please ensure that each figure or table is described clearly, with full legends, and placed at appropriate points.
  • There are numerous grammatical issues, long sentences, unclear phrasing, and inconsistent terminology. A thorough English-language edit is strongly recommended.
  • The conclusion reiterates earlier points but does not offer a forward-looking assessment (e.g., gaps in the field, required preclinical studies, barriers to translation). Please revise to include a more rigorous evaluation of future directions.

Author Response

Reviewer 2, Comment 1: Throughout the manuscript, many figure citations are broken (e.g., Figure 1, 2, 3, etc.). These must be corrected so that readers can follow the structure.

Response:

Thank you for bringing this issue to our attention. The broken figure citations were caused by corrupted cross-reference fields in the previously uploaded Word version. In the revised manuscript, we have fully corrected all figure citations by rebuilding the cross-references and updating the fields throughout the document. All figures are now correctly numbered, properly linked, and accurately cited in the text. We have also rechecked the entire manuscript to ensure that no broken references remain.

Reviewer 2, Comment 2: Several sections repeat the same mechanistic pathways (ROS, NF-κB, JNK, MDM2/p53, EMT). Please reorganize and consolidate overlapping content to improve clarity and flow.

Response:

We sincerely thank you for this insightful and constructive comment. We recognize that the original manuscript contained overlapping descriptions of key signaling pathways (ROS, NF-κB, JNK, MDM2/p53, EMT) across different sections, which could potentially compromise logical flow and readability. We have carefully addressed this issue while preserving the section-specific focus on distinct biological processes (antioxidant effects, anti-inflammation, apoptosis, cell cycle arrest, and metastasis inhibition).

We acknowledge that hinokiflavone (HF) activates or suppresses the same signaling pathways (e.g., ROS, NF-κB, p53) in different cellular contexts to elicit distinct biological outcomes. This context-dependent signaling is a hallmark of natural products with pleiotropic activities. For instance, ROS modulation exhibits dual roles: In stressed hepatocytes, HF acts as an antioxidant to suppress excessive ROS (Section 3.4, lines 473-506); conversely, in tumor cells, HF induces ROS generation to trigger apoptotic cascades (Section 3.1, lines 315-320). NF-κB suppression contributes to multiple effects: anti-inflammatory activity (Section 3.5, lines 527–529 and 594–598), and apoptosis sensitization (Section 3.1, lines 308-318). To maintain clarity while respecting these biological nuances, we opted to retain the current section structure organized by functional outcomes, rather than reorganizing by signaling pathways, which could obscure the context-dependent nature of HF's mechanisms. To improve readability and enhance logical flow, we have implemented the following targeted modifications:

At the end of Section 3.4, we added a short paragraph (lines 513-517) explicitly clarifying HF's dual ROS modulation to prevent reader confusion when encountering ROS-mediated apoptosis in Section 3.1. This addition provides a conceptual bridge between antioxidant and pro-apoptotic activities, eliminating apparent contradiction and enhancing mechanistic coherence: "It is noteworthy that HF exhibits context-dependent ROS modulation: in normal or stressed hepatocytes, HF acts as an antioxidant to suppress excessive ROS accumulation and maintain redox homeostasis; conversely, in tumor cells (as detailed in Section 3.1), HF induces ROS generation to trigger apoptotic cascades, demonstrating its dual regulatory capacity."

At the end of Section 3.5 (lines 618-621), we added a sentence linking NF-κB suppression in the inflammatory context to its role in apoptosis, avoiding redundant explanations while establishing pathway continuity: "Notably, HF's suppression of NF-κB signaling—a master regulator linking chronic inflammation to cancer—also plays a critical role in its direct antitumor cytotoxicity by removing survival signals that counteract apoptosis (detailed mechanisms are elaborated in Section 3.1)."

In the middle of Section 3.1, we added a sentence (lines 269-271) to provide readers with a conceptual framework first, then delivers supporting evidence to improve clarity: “HF induces apoptosis through an integrated multi-layered signaling network. The following subsections delineate these mechanisms in detail, organized by regulatory nodes.”

At the first paragraph of Section 3.2, we added a sentence (lines 333-337), clarifying that cell cycle arrest is a downstream consequence of p53 activation (already detailed in Section 3.1), allowing Section 3.2 to focus on cell type-specific arrest patterns and have a better flow: "As elaborated in Section 3.1, HF activates p53, which functions not only to induce apoptosis but also to enforce cell cycle checkpoints via transcriptional upregulation of cyclin-dependent kinase inhibitors such as p21. This section focuses on the cell type-specific patterns of HF-induced cell cycle arrest and the downstream effector proteins involved.”

Reviewer 2, Comment 3: The manuscript jumps from pharmacodynamics to pharmacokinetics, then to antimicrobial, anti-inflammatory, and anticancer effects without clear transitions. I recommend reorganizing the review into a more cohesive narrative.

Response:

We sincerely thank the reviewer for the insightful suggestion to improve the narrative flow. In response, we have carefully incorporated clear transitional sentences at the beginning of each major section (e.g., preceding Sections 3, 3.1-3.5, 4, and 5) to logically connect the themes of pharmacodynamics, pharmacokinetics, and formulation strategies, thereby creating a more cohesive and reader-friendly narrative throughout the manuscript.

(see the first paragraph of section: 3. Pharmacological Mechanisms of HF's Anticancer Effects; lines 212-217. The first paragraph of section: 3.1. Induction of Apoptosis; lines 245-246. The first paragraph of section: 3.2. Cell Cycle Arrest; lines 329-330. The first paragraph of section: 3.3. Inhibition of tumor metastasis; lines 378-382. The first paragraph of section: 3.4. Antioxidant Effects; lines 420-422. The first paragraph of section: 3.5. Antimicrobial and Anti-inflammatory Effects; lines 523-524. The first paragraph of section: 4. Pharmacokinetic Properties and Formulation Development of HF; lines 669-672.)

Reviewer 2, Comment 4: Many conclusions are drawn from cell-line experiments without sufficient discussion of in vivo data or clinical relevance. Please incorporate more critical evaluation of animal studies and address limitations of in vitro findings.

Response:

We sincerely thank you for this critical and constructive feedback. We recognize that the original manuscript placed disproportionate emphasis on in vitro cell culture data without adequately clarifying which findings had been validated in animal models, potentially creating the misleading impression that all conclusions were based solely on cell line experiments. We have now substantially revised the manuscript to enhance transparency regarding in vivo validation status and to provide rigorous critical evaluation of both in vitro and in vivo evidence.

We recognized that the original Table 2 listed only cell lines and key mechanistic findings without indicating whether these mechanisms had been confirmed in animal models. This omission may have contributed to your concern about over-reliance on cell culture data. Therefore, we added a new column titled "In vivo validation" to Table 2, which provides immediate transparency, allowing readers to quickly assess which mechanistic claims are supported by in vivo evidence and validation experimental methods.

Throughout the manuscript, especially in Section 3, we have systematically revised the text to clearly distinguish between in vitro-only findings and those confirmed in animal models. For mechanisms with in vivo validation, we have added specific details about animal model type or stated as validated both in vitro and/or in vivo. Here are the representative text revisions:

Example 1: Section 3.5  

Revised text (highlighted in vivo validation, lines 578-584): “HF also exerts activity against methicillin-resistant Staphylococcus aureus (MRSA) by inhibiting caseinolytic protease P (ClpP; IC50 34.36 μg/ml) , a virulence factor regulating toxin production and biofilm formation. In a murine MRSA-induced lethal pneumonia model, HF combined with vancomycin significantly improved survival rates (from 60% to 70% compared to vancomycin alone) and reduced lung bacterial burden, demonstrating in vivo validation of ClpP-mediated virulence attenuation and synergistic potential with standard antibiotics[119].”

Example 2: Section 3.1

Revised text (highlighted in vivo validation, lines 301-320): “HF also regulates upstream signaling related to Bcl-2/Bax. The anti-tumor efficacy of hinokiflavone (HF) has been rigorously demonstrated in vivo in colorectal cancer models. In the study by Zhou et al., HF treatment significantly suppressed tumor growth in nude mice bearing HCT116 colorectal cancer xenografts, with dose-dependent reductions in both tumor volume and weight compared to control animals. Mechanistically, HF administration in vivo resulted in increased apoptotic cell counts within tumor tissues, elevated Bax/Bcl-2 ratio, and enhanced caspase-3 activation, directly confirming mitochondrial apoptosis induction at the tissue level. These in vivo results are consistent with in vitro findings that HF induces potent, ROS-mediated apoptosis in colorectal cancer cells through modulation of the JNK and p38 MAPK pathways and inhibition of NF-κB activity[84]. In liver cancer cells, HF dose-dependently induces mitochondrial ROS accumulation, activating JNK and p38 MAPK, and JNK inhibition by SP600125 has been shown to rescue HF-induced apoptosis, restore the Bcl-2/Bax ratio, and downregulate cleaved caspase-3[64]. Similarly, in chronic myelogenous leukemia K562 cells, HF activates the JNK/p38 MAPK axis, suppresses NF-κB, and induces caspase-dependent death. Notably, HF also promotes autophagy, as evidenced by increased LC3-II and decreased p62 expression, with partial rescue by the autophagy inhibitor chloroquine, suggesting HF-induced autophagy may serve a cytoprotective role[68]. Additionally, computational studies predict that HF inhibits PIM1 kinase, an oncogenic serine/threonine kinase, further supporting its pro-apoptotic and anti-tumor potential[20].”

Example 3: Section 3.2

Revised text (highlighted in vivo validation, lines 345-364): “The ability of HF to induce cell cycle arrest has been demonstrated both in vivo and across multiple cancer cell types in vitro, highlighting its broad antiproliferative potential. In hepatocellular carcinoma, Mu et al. showed that HF significantly suppressed tumor growth in SMMC-7721 xenograft-bearing nude mice. Mechanistically, HF administration upregulated phosphorylated p53 (Ser15) and p21 within tumor tissues, while downregulating key cell cycle regulators such as cyclin D1, CDK4, and CDK6, providing in vivo evidence for activation of the p53/p21 axis and G0/G1 phase arrest. These molecular alterations correlated with reduced tumor proliferation (decreased Ki-67 staining) and increased apoptosis (TUNEL positivity) in tumor sections, confirming that cell cycle blockade is an integral part of HF’s antiproliferative action in vivo[64]. In agreement with these animal model findings, in vitro studies further reveal the context-dependent nature of HF-induced cell cycle arrest. In SMMC-7721 and HepG2 hepatocellular carcinoma cell lines, HF exposure resulted in time- and dose-dependent inhibition of viability, G0/G1 phase accumulation, and similar changes in cell cycle protein expression as observed in vivo [64]. In contrast, HF induces cell cycle arrest at distinct checkpoints in other cancer models. G2/M phase accumulation is observed in chronic myelogenous leukemia K562 cells and HCT116 colon cancer cells, with the latter also exhibiting upregulation of p21 and 14-3-3σ and alleviation of MDM2-mediated p53 repression[67, 68]. In melanoma cell lines A375 and B16, HF triggers S phase arrest, as demonstrated by flow cytometry and EdU incorporation assays[63,65].

Reviewer 2, Comment 5: Given that HF is described as a potential therapeutic, a dedicated section discussing safety, toxicity, off-target effects, and pharmacological limitations is necessary.

Response:

We thank the reviewer for this insightful and critical suggestion. We agree that a discussion on safety and toxicological profile is indispensable for a comprehensive evaluation of any potential therapeutic agent. In direct response to this comment, we have now added a dedicated new subsection titled “3.6. Safety, Toxicity, and Pharmacological Limitations” in the revised manuscript (please see pages xx, line xx). This new subsection synthesizes the existing data on HF's selective cytotoxicity towards cancer cells versus normal cells, its documented hepatoprotective effects, and critically discusses the key pharmacological challenges—such as its context-dependent effects on key signaling pathways (e.g., AKT/STAT3). It also explicitly outlines the current knowledge gaps, including the lack of chronic and genotoxicity studies. We believe this addition significantly strengthens the translational perspective of our review and provides a more balanced and critical assessment of HF's clinical potential.

(see section: 3.6. Safety, Toxicity, and Pharmacological Limitations; lines 635-666)

Reviewer 2, Comment 6: The manuscript mentions inconsistent pharmacological results but does not critically evaluate possible causes or implications. A more rigorous comparison of conflicting reports is needed.

Response:

We sincerely thank the reviewer for this insightful comment, which has helped us significantly enhance the critical depth and scholarly impact of our review. We agree that a mere mention of inconsistencies is insufficient. In the revision, we have now added some new content in the "Discussion and Perspectives" chapter, which provides a rigorous comparison of conflicting reports by systematically analyzing the potential causes from three key perspectives: (i) the phytochemical complexity and "matrix effects" from different plant sources, (ii) the inherent limitations of diverse experimental methodologies and disease models, and (iii) the uncertainties regarding compound purity and metabolism. Furthermore, we discuss the implications of these inconsistencies, transforming them from mere contradictions into a roadmap for standardizing future research and understanding HF's context-dependent mechanisms. We believe this addition provides the critical evaluation requested and substantially strengthens the manuscript.

(see the fourth paragraph of section: 5.3. Critical Evaluation of Inconsistent Findings and Limitations in HF Research; lines 941-959)

Reviewer 2, Comment 7: While PK parameters are described, there is little mechanistic or comparative insight. Please discuss why HF bioavailability is low, how metabolism limits therapeutic use, and how different formulations address these barriers.

Response:

We sincerely thank the reviewer for raising this critical point. In response, we have thoroughly revised the pharmacokinetics section to provide the requested mechanistic and comparative insight. The updated text now explicitly discusses the sequential barriers of poor aqueous solubility and extensive pre-systemic metabolism (particularly the cleavage of the critical C-O-C ether bond) as the root causes of HF's low bioavailability, and critically analyzes how different advanced formulations—including solid dispersions, nanomicelles, and metal-organic frameworks—are rationally designed to address these specific limitations in a targeted manner, thereby offering a deeper perspective on formulation strategies to overcome PK challenges.

(see the second paragraph of section: 4. Pharmacokinetic Properties and Formulation Development of HF; lines 675-681. The fourth paragraph of section: 4. Pharmacokinetic Properties and Formulation Development of HF; lines 717-721. The sixth paragraph of section: 4. Pharmacokinetic Properties and Formulation Development of HF; lines 746-747. The seventh paragraph of section: 4. Pharmacokinetic Properties and Formulation Development of HF; lines 764-767. The eighth paragraph of section: 4. Pharmacokinetic Properties and Formulation Development of HF; lines 776-781 )

Reviewer 2, Comment 8: The manuscript lists molecular mechanisms but does not integrate them into a unified model explaining HF’s anticancer effects. Please provide thematic summaries that unify the described pathways.

Response:

We sincerely appreciate this insightful critique. You are absolutely correct that the original manuscript, while comprehensive in detailing individual mechanisms, lacked an integrative framework to elucidate how these pathways converge to produce HF's anticancer effects. We have now added new discussion subsection: "Unified Hierarchical Model of HF's Anticancer Mechanisms and Context Dependence" (Section 5, lines 786-830) in Discussion and Perspectives to provide a unified, hierarchical model that synthesizes HF's molecular targets into a cohesive mechanistic narrative.

Reviewer 2, Comment 9: The antiviral activity is interesting but feels loosely connected to the main cancer-focused narrative. Consider tightening or more clearly linking this section to the manuscript’s main theme.

Response:

We thank the reviewer for this valuable suggestion to better integrate the narrative. In direct response, we have now tightened the connection between the antiviral activity and the main cancer theme in Section 3.5. Specifically, we have added text at the beginning of the antiviral discussion to explicitly state its relevance to cancer chemoprevention by targeting oncogenic viruses. Furthermore, we have added commentary linking the activity against specific viruses (e.g., EBV) directly to their oncogenic potential and have included a concluding paragraph that synthesizes how HF's antimicrobial and anti-inflammatory effects collectively contribute to reshaping the tumor microenvironment and suppressing tumorigenesis, thereby creating a more cohesive and cancer-focused narrative.

(see the second paragraph of section: 3.5. Antimicrobial and Anti-inflammatory Effects; lines 539-541; lines 550-551;lines 558-560; lines 566-573.)

Reviewer 2, Comment 10: The review highlights HF’s benefits but provides little context relative to similar compounds (e.g., amentoflavone) or established anticancer drugs. A comparative discussion would strengthen the manuscript.

Response:

We thank the reviewer for this important and constructive suggestion. In response, we have substantially revised the manuscript to include a dedicated comparative section (Section 5.2, Comparative Analysis of HF with Structurally Related Amentoflavone and Clinical MDM2 Inhibitors, lines 831-889).

This section systematically compares the structural and pharmacological differences between HF and amentoflavone (AF), referencing up-to-date primary and review literature. It provides detailed analysis of their respective in vitro potency (supported by explicit IC50 data), in vivo anticancer efficacy in hepatocellular and pancreatic cancer models, and differences in key molecular mechanisms, including the unique ferroptosis induction and hedgehog/Gli1 regulation for AF, versus CK2 inhibition and splicing modulation for HF. We also integrate a balanced discussion of their pharmacokinetics and metabolism, preclinical development status, and translational barriers.

Furthermore, recognizing the clinical relevance of contemporary targeted therapies, we benchmark HF’s multi-target profile, resistance mechanisms, toxicity, and translational hurdles alongside established clinical trial MDM2 inhibitors, referencing Phase I–III studies and expert reviews. Our discussion emphasizes the necessity of biomarker-driven patient selection, by analogy to classic precision oncology paradigms for EGFR- and BRAF-targeted drugs, as well as the feasibility and clinical rationale of combining HF with other chemotherapeutic or targeted agents.

These comprehensive revisions directly address the reviewer’s concern by clearly situating HF within the landscape of related natural compounds and clinical-stage targeted therapies, and by providing a rigorous, literature-based context that will facilitate understanding and future translational progress.

Reviewer 2, Comment 11: The manuscript describes nanomicelles, MOFs, and amorphous solids but does not provide quantitative improvements (e.g., fold-increase in solubility or bioavailability). Including comparative metrics would improve clarity.

Response:

Thank you for this valuable suggestion. We agree that providing comparative metrics is essential. In the revised manuscript, we have added a new summary table (Table 4) in Section 4 to quantitatively illustrate the improvements achieved by different formulations relative to free HF. While explicit fold-increases in solubility were not consistently quantified across all cited studies, Table 4 focuses on the translational outcome of these formulation strategies: the enhancement of pharmacological activity. Specifically, it quantifies the fold-increases in in vitro cytotoxicity and in vivo antitumor efficacy (e.g., tumor inhibition ratios and apoptosis rates) for nanomicelles and MOFs compared to free HF. These metrics directly reflect the functional benefits of improved solubility and bioavailability.

(see Table 4. Pharmacological activity enhancements of different HF formulations relative to free hinokiflavone; lines 779-781.)

Reviewer 2, Comment 12: Some claims (e.g., “HF demonstrates superior selective cytotoxicity”) are not sufficiently supported by data. Please moderate these claims or provide stronger evidence.

Response:

Thank you for pointing out this important issue. We agree that the previous statement “HF demonstrates superior selective cytotoxicity” may overstate the strength of the current evidence. In the revised manuscript, we have moderated this claim to more accurately reflect the available data and emphasized that existing findings are primarily based on in vitro comparisons between cancer and normal cell lines. Additionally, we now provide supporting IC50 values from reported studies to illustrate the relative differences in cytotoxicity. For example, HF exhibits IC50 values of 4.93–23 μM in several cancer cell lines, whereas the IC50 values in normal human fibroblasts (BJ-FB) and hepatocytes (L02) exceed 50–159 μM (Table 1). This evidence suggests a potential trend of lower cytotoxicity toward normal cells, although more comprehensive in vivo validation is still required. The wording has been updated accordingly in Section 3 of the revised manuscript.

(see the first paragraph of section: 3. Pharmacological Mechanisms of HF's Anticancer Effects; lines 223-226.)

Reviewer 2, Comment 13: Figures are insufficiently referenced or explained in the text. Please ensure that each figure or table is described clearly, with full legends, and placed at appropriate points.

Response:

We appreciate the reviewer’s careful observation. In the revised manuscript, we have ensured that each figure and table is clearly introduced and referenced at its first appearance within the text. Descriptive lead-in sentences have been added to help readers understand the purpose and relevance of each figure.

Reviewer 2, Comment 14: There are numerous grammatical issues, long sentences, unclear phrasing, and inconsistent terminology. A thorough English-language edit is strongly recommended.

Response:

We sincerely appreciate your constructive feedback regarding the language quality of the manuscript. Your point regarding grammatical issues, long sentences, and inconsistent terminology is well-taken, and we have undertaken a comprehensive revision of the entire manuscript. Specifically, we have taken the following actions.

We have systematically reviewed the entire manuscript and addressed the following issues: We corrected grammatical problems, including verb tense consistency and subject-verb agreement (e.g., "enables resistance to death signals and potentially leading to malignant transformation" was revised to "enables resistance to death signals, potentially leading to malignant transformation", lines 258-260). We systematically broke down overly long, multi-clause sentences into shorter, more concise statements throughout the manuscript; for instance, in the Introduction section (lines 54-59), we separated a 60+ word sentence into 2-3 shorter sentences for improved clarity. We standardized terminology and spelling conventions by unifying "tumor" (US English throughout; removed "tumour"), ensuring consistent use of "signaling" vs "signalling" according to American English conventions, standardizing acronyms such that "in vitro" and "in vivo" are consistently italicized, and unifying "cell-cycle" usage based on grammatical context. We also corrected typographical errors (e.g., "intrinsci" to "intrinsic"), removed formatting inconsistencies, and performed multiple comprehensive proofreading passes to ensure accuracy. To ensure the quality of this revision, a native English speaker with experience in biomedical writing reviewed the manuscript, we verified consistency of specialized terminology throughout, and we performed multiple passes to catch subtle grammatical and stylistic issues. Specifically, we have taken the following actions. Specific examples of these revisions are provided below.

The comprehensively edited manuscript is now submitted with these revisions incorporated throughout. We believe the language quality, sentence structure, and terminology consistency have been significantly improved, making the review more accessible and professional.

Reviewer 2, Comment 15: The conclusion reiterates earlier points but does not offer a forward-looking assessment (e.g., gaps in the field, required preclinical studies, barriers to translation). Please revise to include a more rigorous evaluation of future directions.

Response:

We sincerely thank the reviewer for this critical suggestion. We agree that a forward-looking perspective is essential for a comprehensive review. In direct response, we have substantially revised the "Discussion and Perspectives" section to provide a rigorous evaluation of future directions. The revised conclusion now moves beyond a summary to outline specific, critical gaps and required preclinical studies, including: 1) the necessity for in-depth safety and metabolite profiling, 2) the need to resolve context-dependent mechanisms using multi-omics approaches, 3) the strategic development of rational combination therapies to overcome resistance, and 4) the design of next-generation, intelligent delivery systems. We believe this addition provides a clear and actionable roadmap for future research, significantly enhancing the translational impact of our manuscript.

(see the second paragraph of section: 5.4. Clinical Translation Strategies and Future Perspectives; lines 990-1008)

Reviewer 3 Report

Comments and Suggestions for Authors

This review article summarizes the anticancer effects of hinokiflavone. It addresses both its direct anticancer effects and the compound's potential micromodulation of the environment in which cancer develops. The article provides a critical perspective on the discussion and includes attractive illustrations. However, the topic presented in the article has already been addressed in recently published papers. Therefore, the gap in knowledge addressed by the authors should be further clarified and specified. Below, I present my suggestions and reservations regarding the article.

Comment 1: Over the last few years (2021, 2024), review papers focusing on the activity of Hinokiflavone in medicine and cancer therapy have been published.Examples include:Goossens, J. F., Goossens, L., & Bailly, C. (2021). Hinokiflavone and related C–O–C-type biflavonoids as anti-cancer compounds: properties and mechanism of action. Natural Products and Bioprospecting, 11(4), 365-377. https://doi.org/10.1007/s13659-021-00298-w;  Patel, D. K. (2024). Biological potential and therapeutic effectiveness of hinokiflavone in medicine: the effective components of herbal medicines for treatment of cancers and associated complications. Current Nutrition & Food Science, 20(4), 439-449. https://doi.org/10.2174/1573401319666230602121227. Considering that these are also review papers, it would be appropriate to address them in the manuscript (and not only marginally, as was done with reference {18}). The authors should acknowledge that previous reviews have already been published and clearly indicate how the present manuscript adds value — that is, which research gap it explores that was not covered in earlier works.

Comment 2: The methodology is missing in the manuscript. There is no information regarding the framework of the review, i.e., the criteria used by the Authors for selecting the material and studies included in the analysis. The time frame covered, the keywords used, etc., are not provided, nor is it indicated whether any exclusion criteria were applied.

Comment 3: In many places in the manuscript, references are missing – the phrase "Error! Reference source not found.." appears, for instant in lines: 109, 76, 440. This should be changed.

Comment 4: Lines 66-67 – the statement/phrase appeared: “This situation has shifted research focus toward biflavonoids, which exhibit enhanced bioactivity.” However, the statement immediately following this fragment was not supported by any research. This should be rephrased and the source of the information provided.

Comment 5: The article's topic is "Hinokiflavone, a Potential Antitumor Agent: From Pharmacology to Pharmaceutics," meaning the article should focus specifically on HK's activity in the context of cancer. However, sometimes the content deviates from the topic.

Comment 6: Line 71 – the authors refer to the α-glucosidase/n silico docking activity described in the cited article [19], this activity was also analyzed in the article [20], and in [24] the effect on amyloid-β toxicity and fibrillogenesis was studied – how do these specific examples relate to cancer therapy and the current article? Explain, rephrase.

Comment 7: This article primarily addresses cancer action, and that should be the focus. It's worth first presenting the issues underlying direct action against cancer, and only then addressing issues related to indirect action on the microenvironment. I think it's worth reversing the order of the subsections.

Comment 8: The authors present a general section titled “Pharmacological Mechanisms of HF’s Anticancer Effects.” The title implies that this section should focus specifically on HF; however, in the subsection on Antioxidant activity (lines 203–219), the effects of various extracts and different compounds are described. Similarly, in the Antibacterial section (lines 296–297) and Anti-inflammatory section (lines 314–315), the discussion refers to extracts containing HF rather than HF itself. The biological effects of an extract may result from multiple constituents, not only HF. This is not aligned with the stated topic. It was expected that the discussion would focus specifically on HF—the central subject of the review.

This issue occurs in other parts of the manuscript as well. The authors should clearly decide what the focus of the review is: extracts containing HF, or HF as an isolated flavonoid, or other related flavonoids/biflavonoids. I recommend revising the manuscript accordingly to ensure thematic consistency.

Comment 9: Table 1 – Presenting the same types of cells side by side in the table, and including the specific cell lines where possible, would be a better arrangement. In addition, in the “Ref” column, only the reference number should be provided.

Comment 10: The subject of the article is HF, while the content in lines 543-553 of Figure 4 concerns amentoflavone. As previously stated, this is not about HF. The figure is misleading because the caption refers to HF.

Comment 11: Abbreviations used in the article should be explained.

Author Response

Reviewer 3, Comment 1: Over the last few years (2021, 2024), review papers focusing on the activity of Hinokiflavone in medicine and cancer therapy have been published.Examples include:Goossens, J. F., Goossens, L., & Bailly, C. (2021). Hinokiflavone and related C–O–C-type biflavonoids as anti-cancer compounds: properties and mechanism of action. Natural Products and Bioprospecting, 11(4), 365-377. https://doi.org/10.1007/s13659-021-00298-w; Patel, D. K. (2024). Biological potential and therapeutic effectiveness of hinokiflavone in medicine: the effective components of herbal medicines for treatment of cancers and associated complications. Current Nutrition & Food Science, 20(4), 439-449. https://doi.org/10.2174/1573401319666230602121227. Considering that these are also review papers, it would be appropriate to address them in the manuscript (and not only marginally, as was done with reference {18}). The authors should acknowledge that previous reviews have already been published and clearly indicate how the present manuscript adds value — that is, which research gap it explores that was not covered in earlier works.

Response:

We sincerely thank the reviewer for this insightful comment. In response, we have now added a dedicated paragraph in the introduction that explicitly acknowledges the previous reviews by Goossens et al. (2021) and Patel (2024) and clearly positions our manuscript's unique contribution. Our review provides a significant update by synthesizing the most recent discoveries of HF's mechanisms (e.g., MDM2-p53 axis, CK2 inhibition) and distinctly focuses on critically analyzing controversial pharmacological data, providing an in-depth discussion of pharmacokinetic challenges and advanced nanoformulation strategies, and offering a forward-looking perspective on translational barriers—aspects not comprehensively covered in earlier works, thereby establishing our manuscript as an updated and critically focused guide for advancing HF towards clinical application.

(see the fourth paragraph of section: 1. Introduction; lines 115-122 and the last paragraph of section: 1. Introduction; lines 174-184)

Reviewer 3, Comment 2: The methodology is missing in the manuscript. There is no information regarding the framework of the review, i.e., the criteria used by the Authors for selecting the material and studies included in the analysis. The time frame covered, the keywords used, etc., are not provided, nor is it indicated whether any exclusion criteria were applied.

Response:

We sincerely appreciate this suggestion to enhance the rigor of our review. In the revised manuscript, we have added a dedicated new section (Section 2. Methodology) to explicitly outline our literature search strategy. As requested, this section now clearly describes the databases searched, time frame covered, search keywords, inclusion and exclusion criteria, and the screening approach for selecting the studies included in this review. These additions provide transparency regarding the structure and methodological framework of the review.

(see section: 2. Methodology; lines 187-209)

Reviewer 3, Comment 3: In many places in the manuscript, references are missing – the phrase "Error! Reference source not found." appears, for instant in lines: 109, 76, 440. This should be changed.

Response:

Thank you for pointing out this issue. The appearance of “Error! Reference source not found.” was caused by broken cross-reference fields in the previously uploaded version of the manuscript. In the revised version, we have thoroughly corrected all citation errors by rebuilding the cross-references and updating all fields throughout the document. All instances of broken reference links have been removed, and the corresponding citations have been restored to their correct numbering and positions. The entire manuscript has been rechecked to ensure that no remaining reference or figure/table citation errors persist.

Reviewer 3, Comment 4: Lines 66-67 – the statement/phrase appeared: “This situation has shifted research focus toward biflavonoids, which exhibit enhanced bioactivity.” However, the statement immediately following this fragment was not supported by any research. This should be rephrased and the source of the information provided.

Response:

Thank you for pointing out this issue. We agree that the previous wording overstated the evidence and lacked appropriate citations. In the revised manuscript, we have rephrased the statement to moderate the tone and added multiple supporting references. These studies demonstrate that 3'–8''-linked biflavonoids often display enhanced antioxidant activity, stronger enzyme inhibition, and improved cytoprotective or anticancer potential compared with their monomeric flavonoid units. The revised sentence and citations have been incorporated into the Introduction section.

(see the second paragraph of section: 1. Introduction; lines 70-75)

Reviewer 3, Comment 5: The article's topic is "Hinokiflavone, a Potential Antitumor Agent: From Pharmacology to Pharmaceutics," meaning the article should focus specifically on HK's activity in the context of cancer. However, sometimes the content deviates from the topic.

Response:

We sincerely thank you for this important comment. We understand your concern that discussions of HF's antimicrobial, anti-inflammatory, and antioxidant activities, etc. might appear tangential to the anticancer theme. We have carefully revised the manuscript to explicitly establish how these mechanisms are integral to HF's cancer-preventive and cancer-therapeutic potential.

Below are key revisions:

We have strengthened the cancer-relevance framework by emphasizing that: (1) Oncogenic pathogens such as HPV, EBV, and Helicobacter pylori directly drive malignant transformation and are established carcinogenic factors—thus, HF's antimicrobial activity directly contributes to cancer prevention (from lines 529 to 534); (2) Chronic inflammation is a hallmark of cancer development that sustains pro-tumorigenic microenvironments and impairs antitumor immunity—HF's suppression of inflammatory pathways (NF-κB, TNF-α, IL-6) therefore constitutes a cancer-preventive mechanism (from lines 523-534); (3) Oxidative stress is a critical driver of genomic instability and oncogenic activation—HF's dual antioxidant/pro-oxidant capacity (antioxidant in normal tissues, pro-oxidant in tumor cells) represents a sophisticated, cancer-selective mechanism(from lines 420 to 439).

Besides, in Sections 3.4–3.5, we added or refined transitional statements to more clearly relate each mechanism to cancer prevention or therapy where appropriate. Finally, in the Discussion and Perspectives (Section 5.1), we added a “Unified Hierarchical Model of HF's Anticancer Mechanisms and Context” that integrates all of HF's diverse activities—including antimicrobial, anti-inflammatory, and antioxidant effects—into a hierarchical, context-dependent network demonstrating their synergistic convergence on cancer suppression.

We believe these revisions clearly establish that HF's antimicrobial, anti-inflammatory, and antioxidant properties are not peripheral but essential components of its broad-spectrum anticancer profile, all operating within the framework of tumor prevention and microenvironment modulation.

Reviewer 3, Comment 6: Line 71 – the authors refer to the α-glucosidase/n silico docking activity described in the cited article [19], this activity was also analyzed in the article [20], and in [24] the effect on amyloid-β toxicity and fibrillogenesis was studied – how do these specific examples relate to cancer therapy and the current article? Explain, rephrase.

Response:

We sincerely thank you for this insightful comment and for highlighting the need to clarify the relevance of the cited examples to the anticancer focus of our review. You are absolutely correct that references [19], [20], and [24] primarily describe the enhanced bioactivity of biflavonoids in non-oncological contexts (α-glucosidase inhibition, anti-amyloidogenesis). Our intention in citing these studies was to illustrate the fundamental structure-activity relationship (SAR) principle that the dimeric structure of biflavonoids, compared to monomeric flavonoids, confers superior bioactivity—a principle that underpins their enhanced anticancer potency.

To bridge this general principle directly to our review's focus, we had already cited a highly relevant study by Lotfi et al., which provides a specific anticancer example of this "dimer effect". This in silico study confirms that dimeric flavonoids like hinokiflavone exhibit stronger binding affinity to PIM1 kinase—an oncogenic target overexpressed in glioblastoma—than their monomeric counterpart, apigenin.

In direct response to your comment, and to make this logical connection more explicit for the reader, we have revised the relevant sentence in the manuscript. The modified text now explicitly mentions the findings of Lotfi et al. to demonstrate how the general structural advantages of biflavonoids translate directly into superior potential for cancer therapy. Specific changes are made below.

Revised text: "Compared to monomeric flavonoids, the increased number of phenolic hydroxyl groups in biflavonoids enhances enzyme inhibitory activity[19], and the dimeric structure optimizes spatial conformation, improving target binding affinity. This leads to markedly elevated pharmacological potency across various biological contexts, from enzyme inhibition to anticancer activity[20–24]. For instance, a recent in silico study demonstrated that dimeric flavonoids like hinokiflavone exhibit stronger binding than apigenin to PIM1 kinase, an oncogenic target overexpressed in glioblastoma, exemplifying this "dimer effect" in an anticancer setting[22]. This synergistic effect often surpasses the activity increment attributable solely to increased hydroxyl groups."

We believe this revision successfully links the general SAR principle to our specific anticancer focus, and we thank you again for your constructive feedback, which has helped us improve the clarity and impact of our manuscript.

(see the third paragraph of section: 1. Introduction; lines 78-85)

Reviewer 3, Comment 7: This article primarily addresses cancer action, and that should be the focus. It's worth first presenting the issues underlying direct action against cancer, and only then addressing issues related to indirect action on the microenvironment. I think it's worth reversing the order of the subsections.

Response:

We sincerely thank the reviewer for this insightful and constructive suggestion. We completely agree that, given the primary focus of this review on hinokiflavone's anticancer mechanisms, the manuscript should prioritize direct cytotoxic actions before discussing indirect effects on the tumor microenvironment.

In response, we have restructured Section 2 as follows:

Original subsection order:

3.1. Antioxidant Effects (indirect microenvironment modulation)

3.2. Antimicrobial and Anti-inflammatory Effects (indirect microenvironment modulation)

3.3. Induction of Apoptosis (direct anticancer action)

3.4. Cell Cycle Arrest (direct anticancer action)

3.5. Inhibition of Tumor Metastasis (direct anticancer action)

Revised subsection order:

3.1. Induction of Apoptosis (direct anticancer action—prioritized)

3.2. Cell Cycle Arrest (direct anticancer action)

3.3. Inhibition of Tumor Metastasis (direct anticancer action)

3.4. Antioxidant Effects (indirect microenvironment modulation)

3.5. Antimicrobial and Anti-inflammatory Effects (indirect microenvironment modulation)

This restructuring ensures that direct cytotoxic mechanisms—the core anticancer actions of HF—are presented first, followed by microenvironment-modulating activities that complement and enhance overall therapeutic efficacy. This logical flow better aligns with the manuscript's primary focus on cancer therapy.

Additional revisions made to maintain coherence:

  1. Revised the introductory paragraph of Section 3 to emphasize direct anticancer mechanisms before microenvironment modulation, stating: "HF primarily acts directly on tumor cells, precisely interfering with malignant behaviors by regulating apoptosis, cell cycle progression, and metastasis. Additionally, HF modulates the tumor microenvironment..." (lines 218-223)
  2. Updated Figure 3 caption to reflect the new priority order, describing direct cytotoxic actions first, followed by microenvironment effects. (lines 230-244)
  3. Updated all cross-references and internal citations throughout the manuscript to reflect the new subsection numbering

This reorganization significantly strengthens the manuscript's logical flow and thematic focus, ensuring that direct anticancer mechanisms receive the prominence they merit. We deeply appreciate the reviewer's guidance, which has substantially improved the manuscript's clarity and coherence.

Reviewer 3, Comment 8: The authors present a general section titled “Pharmacological Mechanisms of HF’s Anticancer Effects.” The title implies that this section should focus specifically on HF; however, in the subsection on Antioxidant activity (lines 203–219), the effects of various extracts and different compounds are described. Similarly, in the Antibacterial section (lines 296–297) and Anti-inflammatory section (lines 314–315), the discussion refers to extracts containing HF rather than HF itself. The biological effects of an extract may result from multiple constituents, not only HF. This is not aligned with the stated topic. It was expected that the discussion would focus specifically on HF—the central subject of the review. This issue occurs in other parts of the manuscript as well. The authors should clearly decide what the focus of the review is: extracts containing HF, or HF as an isolated flavonoid, or other related flavonoids/biflavonoids. I recommend revising the manuscript accordingly to ensure thematic consistency.

Response:

We sincerely appreciate the reviewer's critical and constructive observation regarding thematic consistency. The reviewer has identified a fundamental issue that substantially weakens the manuscript: the original text inappropriately conflated data from crude plant extracts containing HF with evidence from purified/isolated HF studies, creating confusion regarding the actual focus of the review. We fully agree with this assessment and have undertaken comprehensive revisions to prioritize and clearly distinguish between purified HF studies and extract-guided evidence. Specifically, we have removed all references to crude extract studies where HF's specific contribution could not be delineated. Retained only scientifically justified extract-derived data where HF is documented as the predominant active constituent, and where such data is appropriately contextualized with caveats regarding the need for bioassay-guided fractionation. For example, we have systematically addressed this issue at all three locations identified by the reviewer:

  1. Section 3.4 (Antioxidant Effects, lines 453–471): “Beyond dietary antioxidants, specific natural compounds such as HF have demonstrated intrinsic free radical-scavenging capacity. Using DPPH-UPLC-Q-TOF/MS screening, Wang et al. demonstrated that purified HF exhibited the most potent radical-scavenging activity among biflavonoids tested, including podocarpusflavone A, bilobetin, and ginkgetin[62]. Target‑guided isolation by offline DPPH‑HPLC followed by high‑speed countercurrent chromatography has corroborated these screening results and yielded purified HF for direct bioassay; in those targeted assays HF displayed measurable DPPH scavenging (reported at 75 μM), whereas small‑molecule monoflavonoids such as quercetin exhibited lower IC50 values in the same screening framework (e.g., quercetin IC50 ≈ 3.2 μM)[99]. Critically, HF's antioxidant effects extend beyond in vitro radical scavenging to encompass redox-modulating activity in biological systems. In a high-fat diet-induced rat model, purified HF (25–50 mg/kg) significantly attenuated oxidative stress by activating the EGFR/PI3K/Akt/eNOS signaling axis, enhancing nitric oxide bioavailability and upregulating SOD1 and catalase expression, collectively reducing hepatic ROS and lipid peroxidation[100]. These complementary lines of evidence-(i) identification of HF as a dominant radical scavenger and (ii) in vivo attenuation of oxidative stress through redox-modulating signaling—establish that HF possesses biologically and therapeutically relevant antioxidant activity in the context of chronic disease prevention and TME modulation.”
  2. Section 3.5 (Antimicrobial and Anti-inflammatory Effects, lines 599–605; lines 575-579 )

We acknowledge that this represents extract-guided rather than purified-compound evidence (lines 599-605). However, we believe it merits inclusion for the following reasons: HF is the predominant biflavonoid constituent in Juniperus rigida extracts; No alternative in vivo evidence currently exists in the literature documenting purified HF's anti-inflammatory activity in animal models; Extract-guided identification is an accepted methodology in natural products research, and this evidence provides important translational context; We have explicitly acknowledged the limitation by stating that bioassay-guided fractionation is required to definitively establish HF's specific contribution. Besides we have clarified that Negm et al. isolated and characterized purified HF, but not just extract-level findings, with MICs specifically attributable to purified HF (lines 575-579): “Negm et al. isolated purified HF and structural derivatives from Cycas thouarsii, identifying potent antibacterial activity specifically attributable to purified HF—rather than extract mixtures—against K. pneumoniae clinical isolates, with minimum inhibitory concentrations (MICs) of 0.25–0.5 μg/ml”.

Reviewer 3, Comment 9: Table 1 – Presenting the same types of cells side by side in the table, and including the specific cell lines where possible, would be a better arrangement. In addition, in the “Ref” column, only the reference number should be provided.

Response:

Thank you for this helpful suggestion. In the revised manuscript, we have reorganized Table 1 to group the same categories of cells together, ensuring that specific cell lines are clearly listed to facilitate comparison across studies. Additionally, the “Ref” column has been standardized to display only the reference numbers, in accordance with the reviewer’s recommendation. The updated version of Table 1 has been incorporated into the revised manuscript.

(see Table 1. Summary of reported cytotoxicity (IC50) of HF against different cell lines; lines 377)

Reviewer 3, Comment 10: The subject of the article is HF, while the content in lines 543-553 of Figure 4 concerns amentoflavone. As previously stated, this is not about HF. The figure is misleading because the caption refers to HF.

Response:

We thank the reviewer for this valid point. Our original intention was to introduce the TPGS/Soluplus® mixed micelle system as a proven platform for biflavonoids by citing the study on Amentoflavone (a structural analog of HF) as a precedent. However, we agree that including detailed experimental data (such as specific particle size and cytotoxicity values) for Amentoflavone in a section dedicated to HF is distracting and potentially misleading. In the revised manuscript, we have condensed this description. We now briefly mention Amentoflavone only to establish the utility of the carrier platform for biflavonoids, and then immediately focus on the specific application and results of HF-loaded micelles by Chen et al. [127]. This ensures the narrative remains strictly focused on HF.

(see the seventh paragraph of section 4. Pharmacokinetic Properties and Formulation Development of HF; lines 757-759.)

Reviewer 3, Comment 11: Abbreviations used in the article should be explained.

Response:

We sincerely appreciate the reviewer’s attention to detail regarding abbreviations. We have standardized the abbreviations in the previous version.

In the revised manuscript, rather than adding a separate list, we have conducted a comprehensive in-text standardization to ensure maximum flow and readability:

  1. We have verified that all abbreviations are defined in full upon their first appearance in the text.
  2. We have removed abbreviations that appear only once or very infrequently, using the full term instead to reduce unnecessary cognitive burden on the reader.
  3. Once defined, we have ensured the abbreviation is used consistently throughout the remainder of the manuscript.

We believe these corrections significantly improve the clarity and professional presentation of the review.

Round 2

Reviewer 2 Report

Comments and Suggestions for Authors

The revised manuscript is substantially improved, and I appreciate the careful, point-by-point responses to the previous review.

Best,

Reviewer 3 Report

Comments and Suggestions for Authors

Thank you to the Authors for the explanations. The manuscript has been sufficiently improved. In my opinion, it can be accepted for publication in its current form.